# MINIAPPBENCH: Evaluating the Shift from Text to Interactive HTML Responses in LLM-Powered Assistants

Zuhao Zhang [* 1 2]  Chengyue Yu [* 1]  Yuante Li [* 3]  Chenyi Zhuang [1]  Linjian Mo [1]  Shuai Li [2]

## Abstract

With the rapid advancement of Large Language Models (LLMs) in code generation, human-AI interaction is evolving from static text responses to dynamic, interactive HTML-based applications, which we term **MINIAPPS**. These applications require models to not only render visual interfaces but also construct customized interaction logic that adheres to real-world principles. However, existing benchmarks primarily focus on algorithmic correctness or static layout reconstruction, failing to capture the capabilities required for this new paradigm. To address this gap, we introduce **MINIAPPBENCH**, the first comprehensive benchmark designed to evaluate principle-driven, interactive application generation. Sourced from a real-world application with **10M+** generations, MINIAPPBENCH distills 500 tasks across six domains (e.g., Games, Science, and Tools). Furthermore, to tackle the challenge of evaluating open-ended interactions where no single ground truth exists, we propose **MINIAPPEVAL**, an agentic evaluation framework. Leveraging browser automation, it performs human-like exploratory testing to systematically assess applications across three dimensions: Intention, Static, and Dynamic. Our experiments reveal that current LLMs still face significant challenges in generating high-quality MINIAPPS, while MINIAPPEVAL demonstrates high alignment with human judgment, establishing a reliable standard for future research. Our homepage is available in miniappbench.github.io.

## 1 Introduction

With the rapid advancement of Large Language Models (LLMs) in code generation (Novikov et al., 2025;

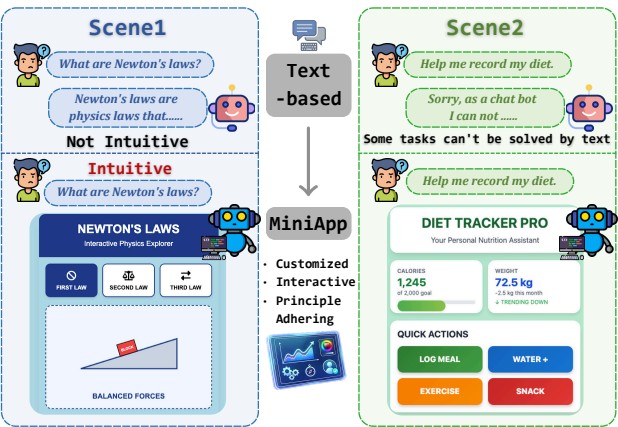

*Figure 1.* **The shift from text to MINIAPPS.** Unlike static text, MINIAPPS transforms abstract explanations into intuitive visualizations and unlocks actionable tasks (e.g., diet tracking) that were previously impossible.

Li et al., 2025c; Xia et al., 2025), models are evolving to `Autonomous Architects` capable of constructing complete software solutions. In this emerging landscape, code transcends its role as a mere intermediate symbolic representation; it becomes a direct executable medium through which a model's internal knowledge is externalized into dynamic, user-facing artifacts. This transformation facilitates a paradigm shift in human-LLM interaction (as illustrated in Figure 1), moving from static text-only responses to rich, code-based engagements. Users now expect LLMs to produce interactive visualizations or functional applications that embody real-world logic. Consequently, to ensure these interactions feel natural and seamless, the model must actively **capture** and **construct** implicit assumptions or principles, such as "an object in free fall follows Newton's laws" or "a week has seven days", which, while often taken for granted in human communication, are essential for valid execution. Real-world cases are shown in Figure 2.

We argue that the web provides a particularly effective substrate for realizing such interactions. In this context, `HTML` represents world states and structural relationships, `CSS` determines perceptual salience, and `JavaScript` encodes causal dependencies, temporal evolution, and interaction logic—together forming an executable world model. Moreover, its interactivity adds an additional layer of depth to this interaction.

*Equal contribution [1]Inclusion AI, Ant Group, Hangzhou, Zhejiang, China [2]Shanghai Jiao Tong University, Shanghai, China [3]Carnegie Mellon University, Pittsburgh, PA, USA. Correspondence to: Chenyi Zhuang <chenyi.zcy@antgroup.com>.

*Proceedings of the 43rd International Conference on Machine Learning*, Seoul, South Korea. PMLR 306, 2026. Copyright 2026 by the author(s).

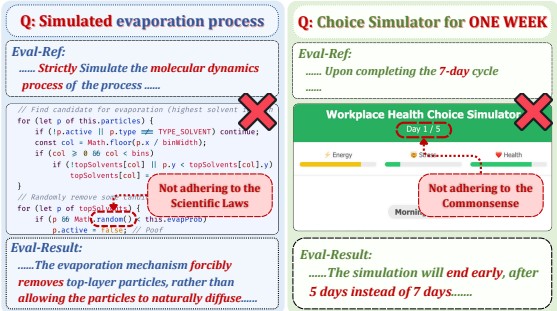

*Figure 2.* **Failure Cases in Principle Adherence.** MINIAPPS require models to capture and instantiate relevant real-world principles, while MINIAPPEVAL proves effective due to its multi-component system design (eval-ref, code, playwright).

From this perspective, we posit that *rendered HTML responses* will emerge as a new form of human–LLM interaction, which we term **MINIAPPS**. Unlike traditional web pages, which primarily focus on static content display or predefined CRUD (Create, Read, Update, and Delete) workflows, MINIAPPS are characterized by two core properties: ❶ **Fidelity to Real-World Principles**, where the model must **capture** and **construct** the implicit principles embedded in the user's query; and ❷ **Customized Interaction**, where application structure and behavior are dynamically synthesized to match user intent, rather than being instantiated from fixed templates.

However, current benchmarks remain tethered to the static past, failing to capture this shift. Traditional code benchmarks like MBPP (Austin et al., 2021) and HumanEval (Chen et al., 2021) focus on algorithmic syntax, treating code as abstract logic divorced from execution context. Conversely, web generation benchmarks (Sun et al., 2025; Lu et al., 2026; Xu et al., 2025) prioritize visual fidelity or static layout reconstruction. This creates a critical blind spot: existing metrics are unable to verify whether LLMs truly capture and construct the underlying real-world principles implied by user queries.

In practice, achieving these properties is non-trivial. As shown in Figure 2, an artifact may be *syntactically valid* and *successfully executable*, but still fail to support high-fidelity, non-fragmented interaction aligned with real user reasoning. To bridge this gap, we introduce **MINIAPPBENCH**, the first benchmark designed specifically to evaluate the ability of LLMs to generate MINIAPPS. Table 1 compares MINIAPPBENCH with representative prior benchmarks. MINIAPPBENCH is constructed through a rigorous multi-stage pipeline that distills tens of millions of real-world user queries into a balanced set of principle-driven, interaction-intensive tasks.

Evaluating **MINIAPPS** also poses a unique challenge due to the inherently *open-ended* nature of application generation. Given that multiple implementations with different structures, interaction patterns, and design choices may all validly satisfy the same user intent, there is often no single canonical "ground truth" code solution.

To address this challenge, we propose a novel *Agentic Evaluation Framework*, **MINIAPPEVAL**. Instead of relying on rigid assertions or template-based matching, MINIAPPEVAL leverages Playwright (Microsoft, 2026) to perform human-like exploratory testing by simulating interactions such as clicking, dragging, and observing runtime behavior. It dynamically verifies the generated application along three complementary dimensions: *Intention*, *Static*, and *Dynamic*. Together, these dimensions assess whether the application fulfills the user's intent, exhibits a coherent static implementation, and demonstrates interactive behavior that adheres to implicit real-world constraints and interaction expectations.

Our main contributions are summarized as follows:

- We rethink the future of human-LLM interaction and argue that **rendered HTML responses** constitute a new interaction paradigm in the form of **MINIAPPS**.

- We propose **MINIAPPBENCH**, the first benchmark dedicated to evaluating principle-driven, interactive application generation. Derived from real-world user demands, it comprises 500 rigorous tasks that challenge LLMs to align executable code with implicit user reasoning.

- We introduce **MINIAPPEVAL**, a novel agentic framework that integrates static inspection with human-like dynamic exploration to holistically assess application fidelity across Intention, Static, and Dynamic.

- Experiments reveal that current LLMs still struggle to reliably construct MINIAPPS, while MINIAPPEVAL achieves high consistency with human judgment, enabling more faithful assessment of next-generation interactive systems.

## 2 Related Work

### 2.1 Code Generation and World Reasoning

Existing code generation benchmarks (Paul et al., 2024; Jiang et al., 2026) have largely focused on assessing functional correctness within the domains of algorithmic logic, software engineering, and data science. Early benchmarks such as HumanEval (Chen et al., 2021) and MBPP (Austin et al., 2021) assess function-level algorithmic reasoning, while more recent efforts like SWE-bench (Jimenez et al., 2024) and MLE-bench (Chan et al., 2025) extend evaluation to repository-scale software maintenance and engineering workflows. Despite this progression in scale and realism, these benchmarks largely treat code as an abstract symbolic artifact whose quality is determined by test passing or task completion. Interaction and user-facing behavior are either absent or tightly constrained by fixed assertions. As a result, they do not capture whether models can use code as an *interactive medium* to externalize knowledge, reason about

real-world principles, or support customized human-LLM interaction—capabilities that are central to **MINIAPPS**.

Conversely, a parallel line of research evaluates LLMs on their understanding of real-world principles. Benchmarks such as PIQA (Bisk et al., 2020) and GSM8K (Cobbe et al., 2021) assess this capability through passive textual inference, asking models to predict outcomes based on described scenarios. In the domain of embodied AI, frameworks like AlfWorld (Shridhar et al., 2021) and Voyager (Wang et al., 2024a) test agents' ability to act within predefined, immutable environments. While these benchmarks explicitly evaluate models' understanding of *explicit* real-world principles within constrained scenarios, they do not assess the ability of models to capture and integrate *implicit* principles and express them through executable artifacts.

### 2.2 Web Development

Early work on web generation (Li et al., 2025b; Ning et al., 2025) mainly focused on visual-to-code translation and static layout reconstruction. Pioneering works like Pix2Code (Beltramelli, 2018) and Web2Code (Yun et al., 2024) treated web generation as an image captioning or translation task, focusing on pixel-level fidelity and structural alignment with reference designs. Similarly, benchmarks like FullFront (Sun et al., 2025) emphasize the visual consistency of the generated frontend. Sketch2Code (Li et al., 2025b) further extended this to hand-drawn sketches. These approaches largely focus on visual appearance, with limited attention to the dynamic logic and state transitions that characterize modern interactive applications. More recent benchmarks have advanced towards Engineering-level Web Development, addressing multi-step or multi-file generation. Frameworks such as WebGenBench (Lu et al., 2026) and WebBench (Xu et al., 2025) evaluate the ability to construct complex file structures for traditional applications like e-commerce sites or forums. However, despite increased structural complexity, these tasks remain centered on information presentation and standard CRUD workflows, often relying on templates and established patterns, with limited need for reasoning about custom interaction rules.

### 2.3 Evaluation Methodologies

Traditional web evaluation paradigms typically rely on static code analysis, visual similarity metrics (e.g., screenshot comparison), or predefined interaction scripts. Approaches like Pix2Code (Beltramelli, 2018) and Web2Code (Yun et al., 2024) adopt snapshot-based evaluation, which captures layout fidelity but overlooks the interaction process. ArtifactsBench (Zhang et al., 2025), on the other hand, analyzes the interaction process through multiple screenshots. Similarly, methods relying on fixed click-scripts, such as WebBench (Xu et al., 2025), FullFront (Sun et al., 2025), cover only narrow, pre-determined paths. In contrast, modern interactive applications feature rich interactivity and

effectively unbounded state spaces. Fixed scripts cannot adapt to diverse valid behaviors or open-ended interaction trajectories implemented by a model. Consequently, static or scripted methods are ill-equipped to evaluate whether a generated application truly functions as a consistent dynamic system.

While recent works have introduced agent-based evaluators (Wang et al., 2024b; Gao et al., 2024) to address interactivity, they predominantly rely on comparative analysis. Systems like WebDevJudge (Li et al., 2025a) and FronTalk (Wu et al., 2025) evaluate quality by measuring deviation from a reference implementation (ground truth) or by performing pairwise preference rankings (A/B testing). Such reference-dependent evaluation is ill-suited for MINIAPPS, where customized and open-ended generation admits multiple equally valid realizations.

## 3 MINIAPPBENCH

### 3.1 Overview

We present **MINIAPPBENCH**, a benchmark comprising 500 tasks designed to evaluate LLMs on their ability to develop **MINIAPPS** as a new form of human-LLM interaction. Moving beyond static layouts or standard CRUD operations found in prior work (Xu et al., 2025; Zhang et al., 2025), our benchmark focuses on **adherence to real-world principles** and **customized interaction**. The dataset is distilled from **tens of millions of real user queries** collected from a large-scale production platform. Through a multi-stage filtration process involving model-based difficulty assessment and manual verification (detailed in Appendix A), we selected 500 high-value queries that span six diverse domains (see Figure 3(e)). Critically, these tasks require models not only to generate syntactically valid code, but also to **construct interactive behaviors that align with user intent by correctly capturing and operationalizing implicit real-world principles**, thereby enabling coherent, natural, and non-fragmented user interactions. The overview of MINIAPPBENCH is provided in Figure 3.

### 3.2 Data Representation

To facilitate structured evaluation and fine-grained analysis, we organize the dataset into a canonical tuple representation. Formally, the dataset is defined as $\mathcal{D} = \{\tau_i\}_{i=1}^N$, where each entry $\tau_i$ is encapsulated as:

$$\tau_i = \langle q_i, (c_i, s_i), r_i, d_i \rangle \tag{1}$$

Here, the components are defined as follows (the data format is described in Appendix A.5):

- $q_i$ represents the **natural-language query** sourced from real users, serving as the input for the model.

- $(c_i, s_i)$ denotes the **two-level taxonomy**, where $c_i \in \mathcal{C}$ is the coarse-grained domain (e.g., Science, Games)

*Table 1.* Comparison of representative benchmarks across three families: code generation, real-world reasoning, and web development. **Real-User** indicates whether queries are sourced from real users. **Div.** (task diversity) is bucketed by the number of primary task categories (Low: $< 3$, Mid: 3–5, High: $> 5$). **Comp.** (task complexity) is approximated by the number of steps in the evaluation protocol (Low: 1, Mid: 2–5, High: $> 5$). **RW-Prin.** indicates whether solving the queries requires real-world principles (e.g., physics or commonsense); details are provided in Appendix A.4.2.

| Benchmark | #Data | Task | Real-User | Div. | Comp. | RW-Prin. |
|---|---|---|---|---|---|---|
| MBPP | 500 | Algorithmic Problem Solving | ✗ | Low | High | Low |
| HumanEval | 164 | Algorithmic Problem Solving | ✗ | Low | High | Low |
| SWE-Bench | 2,294 | Repository-level Bug Fixing | ✗ | High | High | Low |
| MLE-Bench | 75 | Repository-level Software Engineering | ✓ | High | High | Low |
| PIQA | 2,000 | Physical Reasoning | ✗ | Low | Low | High |
| GSM8K | 1,000 | Mathematical Reasoning | ✗ | Low | Low | High |
| AlfBench | 3,553 | Embodied Reasoning | ✗ | Low | Low | High |
| Voyager | N/A | Embodied Reasoning | ✗ | Low | Low | High |
| Pix2Code | 5,250 | Web Interface Cloning | ✗ | Low | Low | Low |
| Web2Code | 1,198 | Web Interface Cloning | ✗ | Low | Low | Low |
| FullFront | 50 | Web Interface Cloning | ✗ | Low | High | Low |
| WebGenBench | 101 | Multi-file Web Dev | ✗ | Mid | High | Low |
| A11YN | 300 | Web Accessibility | ✓ | High | Low | Low |
| WebBench | 50 | Multi-step Iterative Dev | ✗ | High | High | Low |
| FronTalk | 100 | Multi-step Iterative Dev | ✗ | Low | High | Low |
| ArtifactsBench | 1,825 | Interactive Visual Artifacts Dev | ✗ | High | Mid | Mid |
| WebDevArena | N/A | Web Preference (A/B) | ✓ | High | – | – |
| WebDevJudge | 654 | Web Preference (A/B) | ✓ | Mid | High | Mid |
| MINIAPPBENCH | 500 | Customized MINIAPPS Dev | ✓ | High | High | High |

and $s_i$ is the specific subclass, enabling domain-specific performance breakdown.

- $r_i$ is the **structured evaluation reference**. Unlike traditional benchmarks that rely on fixed test cases, $r_i$ specifies verifiable constraints across Intention, Static, and Dynamic dimensions to guide the agentic evaluator.
- $d_i \in \{\text{Easy}, \text{Mid}, \text{Hard}\}$ labels the **task difficulty**, derived from the pass rates of baseline models.

This structured representation supports the open-ended nature of MINIAPPS: the evaluation reference $r_i$ functions as a flexible inspection guide rather than a rigid template, validating any generated artifact that functionally satisfies the user intent $q_i$.

### 3.3 Evaluation Dimension

We design three dimensions to assess the quality of MINIAPPS, comprehensively verify whether the generated application adheres to the real-world principles and interaction expectations specified by the user.

**Intention Dimension.** This score measures whether the MiniApp correctly interprets and fulfills the high-level user goal specified in $q_i$. For example, if the query requests a physics simulation of pendulum motion, the evaluator

checks whether the core dynamics (periodicity, energy conservation) are meaningfully represented.

**Static Dimension.** This score evaluates structural and syntactic correctness without execution. It verifies the presence of required elements, proper code organization, and adherence to accessibility standards. For instance, a weather dashboard should include clearly labeled temperature, humidity, and location fields, despite interaction.

**Dynamic Dimension.** This score evaluates the MiniApp's runtime behavior through multi-step interaction trajectories. It evaluates two critical aspects: (1) Sequential Logic and Planning: The evaluator executes complex chains of actions (e.g., add a new task → mark as complete → verify removal from the active list) to verify that state transitions remain consistent and reversible, faithfully reflecting causal dependencies in the real world. (2) Robustness and Boundary Handling: MINIAPPEVAL is tested against adversarial or edge-case inputs (e.g., submitting an empty string as a task name or inputting invalid dates in a scheduler) to ensure the application handles exceptions gracefully without crashing or violating real-world principles.

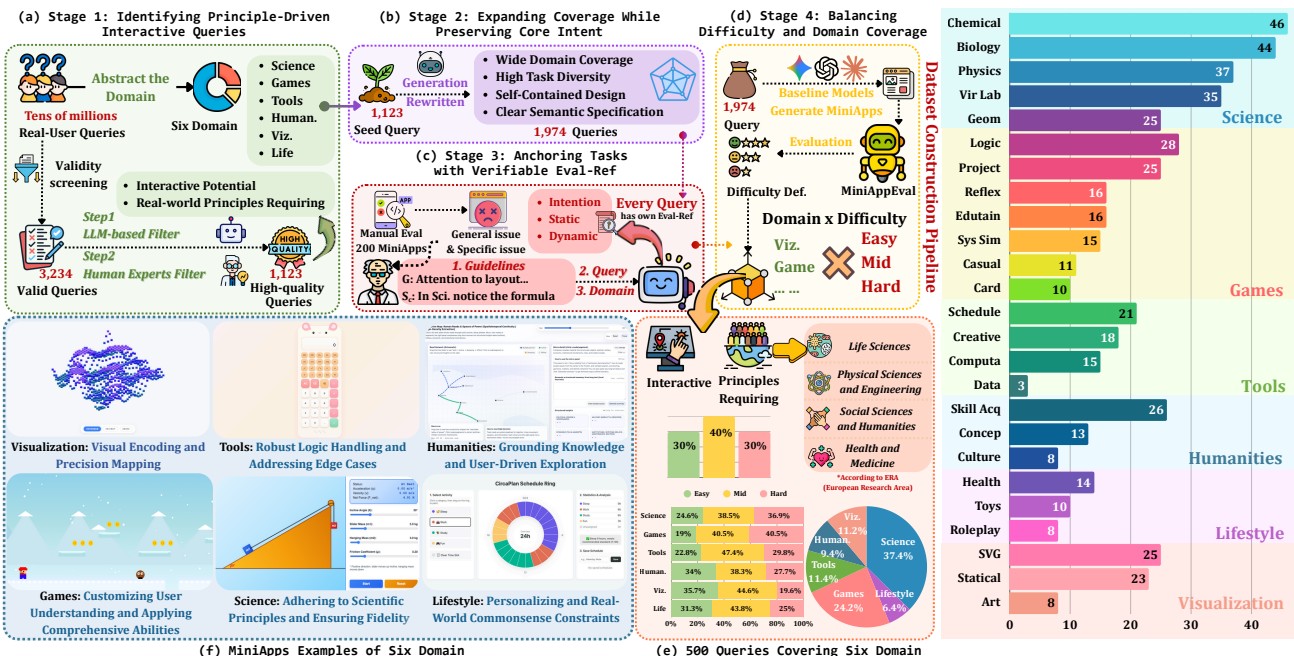

*Figure 3.* **Overview of the MINIAPPBENCH dataset and construction process.** (a)–(d) illustrate the dataset construction pipeline. (e) summarizes the dataset features and distributions (domain and difficulty), with the distribution of subclasses shown in the side bar charts. (f) presents representative MINIAPPS examples from six domains.

## 3.4 Dataset Construction Pipeline

⇛ **Stage 1: Identifying Principle-Driven Interactive Queries.** The first stage tackles a key challenge: *not all real user queries are suitable for evaluating customized interaction or the construction of real-world principles.* Many queries are purely informational, underspecified, or trivially solvable without meaningful interaction logic.

We began with an initial pool of tens of millions of real user queries, from which we sampled a subset and removed invalid entries (e.g., incoherent text, multi-turn follow-ups), resulting in 3,234 candidates. We then used a *LLM-based categorization* approach to group queries by their underlying themes and suitability for interactive tasks. *Human experts* further refined these categories into 6 coarse-grained domains and 25 fine-grained subclasses, ensuring semantic consistency and balanced coverage across knowledge areas (details in Appendix A.1). To ensure data quality, we applied a *hybrid quality filtering strategy*. First, an LLM-driven filter removed queries that were vague, static, or lacking in interactive potential. Second, a manual verification step confirmed that the underlying **principles** and **interactive logic** of each task could **be explicitly materialized through HTML** (the full pipeline is provided in Appendix A.4.2). This rigorous verification ensures that every task in the dataset is suitable for testing the core aspects of the benchmark.

To quantify agreement in this filtering stage, three LLMs (GPT-5.2, Gemini-3-Pro-Preview, and Claude-Sonnet-4.5)

independently assigned binary removal labels. Only unanimously flagged items were discarded, leaving 1,521 queries, with Fleiss' $\kappa = 0.75$. Three human experts then re-screened these queries under the same protocol, retaining 1,123 items, with Fleiss' $\kappa = 0.87$. Both scores indicate substantial agreement.

This stage resulted in 1,123 high-quality seed queries, forming the foundation of the benchmark. These queries are rich in real-world principles and support meaningful evaluation of customized interactions and principle-based generation.

⇛ **Stage 2: Expanding Coverage While Preserving Core Intent.** While the filtered seed queries are high quality, they alone do not provide sufficient coverage of interaction patterns or domain diversity. The second stage therefore focuses on expanding task diversity without diluting the underlying principles. We employ the seed queries as anchors in an LLM-driven evolutionary augmentation process to synthesize variants. These variants explore diverse scenarios, parameter configurations, and interaction structures while strictly maintaining the original intent. Both seed and generated queries then undergo a standardization step, in which they are rewritten to be self-contained, explicit, and engineering-feasible.

This step is critical, as it ensures the benchmark evaluates application construction ability rather than ambiguity resolution or prompt interpretation. After augmentation and standardization, the query set expands to 1,974 candidates. In the final benchmark, 77 tasks (15.4%) are directly drawn

from real online user queries, while 423 tasks (84.6%) are LLM-augmented variants. All augmented tasks are grounded in genuine user seeds and preserve the original topic and intent, while expanding difficulty and domain coverage (Appendix A.3).

➠ **Stage 3: Anchoring Tasks with Verifiable Evaluation References.** We sampled 200 queries from Stage 2 and asked different models to generate MINIAPPS for manual assessment. During this process, we identified both cross-domain issues and domain-specific pitfalls. To enhance the evaluation capability of **MINIAPPEVAL**, we construct evaluation references via a human-guided generation strategy. Specifically, human experts write (i) a set of general guidelines $G$ and (ii) domain-specific instructions $S_{c_i}$ to guide an LLM in generating these references.

Given the query $q_i$, its domain $c_i$, and the guidelines $(G, S_{c_i})$, the LLM maps key evaluation points onto three dimensions aligned with our evaluation dimension and produces a query-specific reference:

$$f_{\text{ref}}(q_i, c_i, G, S_{c_i}) \to r_i. \qquad (2)$$

These references assist the evaluator but are not used as the final decision criterion. We further asked domain experts to audit the generated reference. Their review suggests that the reference effectively surfaces implicit underlying principles that the MINIAPPS generation model might otherwise overlook (Figure 2). Importantly, the references are **not manually refined**, ensuring scalability, generalizability, and full reproducibility.

➠ **Stage 4: Balancing Difficulty and Domain Coverage.** The final stage constructs a balanced, challenging, and statistically meaningful evaluation benchmark. Tasks are assessed along the Intention, Static, and Dynamic dimensions and categorized into Easy, Medium, or Hard levels. To ensure diversity and fairness, we perform stratified sampling, selecting 500 tasks from a combination of domains and difficulty levels, guaranteeing a representative mix. Additionally, we manually review each query before inclusion to ensure the properties of seed queries from Stage 1 are accurately preserved during the expansion process.

The resulting dataset follows a balanced difficulty distribution of 30% Easy, 40% Medium, and 30% Hard, facilitating fair cross-model comparisons while maintaining both challenge and diversity (details in Appendix A.3). It also upholds essential characteristics like implicit principles that can be concretely expressed through HTML and customized interaction.

## 4 Agentic Evaluation Methodology

As discussed in Section 2, assessing only static code or post-execution screenshots fails to verify interface behavior under real user interaction, nor to capture the implicit real-world

principles required by the user's query, which constitute two key challenges in generating high-quality MINIAPPS.

To address these challenges, **MINIAPPEVAL** adopts an **agentic evaluation framework** with dynamic interaction enabled by **browser automation** (Playwright (Microsoft, 2026)). An LLM-powered agent actively interacts with the MiniApp and records the full interaction trajectory. Then based on this trajectory, MINIAPPEVAL produces structured scores along three dimensions: Intention, Static, and Dynamic. Meanwhile, the evaluation framework is designed to **minimize user cost**. Users only need to provide an OpenAI-compatible chat API and can launch the entire evaluation with a single command (details in Appendix B.7; the cost analysis is provided in the Appendix B.10). For each query, the pipeline runs automatically, including code generation and scoring, which helps reduce the impact of extraneous factors unrelated to the model's capabilities.

Overall, our methodology consists of two tightly coupled components: (i) a standardized code generation scaffold and (ii) an LLM-powered autonomous agentic evaluation framework.

### 4.1 Standardized Code Generation Scaffold

We provide an easy-to-use code generation scaffold. This part consists of two stages: **Generation** and **Compilation**.

**Generation.** In the generation stage, the model receives a user query $q_i$ and generates a single, self-contained `index.html` file that integrates the document markup, embedded styling, and functional logic. Our evaluation uses the HTML format, while a standardized React option is also provided for users. The specific system prompt (generation prompt) templates are provided in Appendix D.1.

**Compilation.** In the compilation stage, the generated source code is assembled and validated into a deployable artifact. All artifacts must be self-contained and runnable in a browser without external build tools, network access, or server-side dependencies. To ensure fair comparison, we run them in a standardized Chromium (Playwright) sandbox with fixed runtime conditions and strict isolation, evaluating each artifact independently.

### 4.2 Autonomous Agentic Evaluation Framework

**Input.** The evaluation agent receives four inputs: (i) the original user query $q_i$, (ii) the evaluation reference $r_i$, (iii) the complete generated source code, and (iv) a live, interactable instance of the MiniApp running in the browser. Any natural-language explanations generated by the code model are retained as auxiliary context.

**Evidence Collection.** MINIAPPEVAL uses **Playwright** to simulate a human evaluator: it loads the generated MiniApp, observes its initial state, and autonomously interacts with it based on the user query $q_i$. All interactions (clicking/typing)

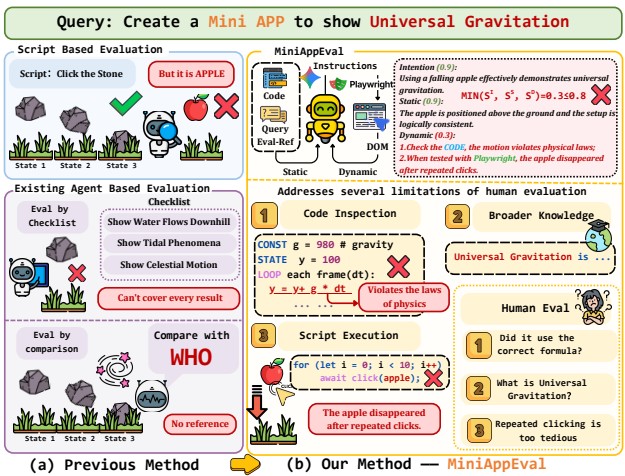

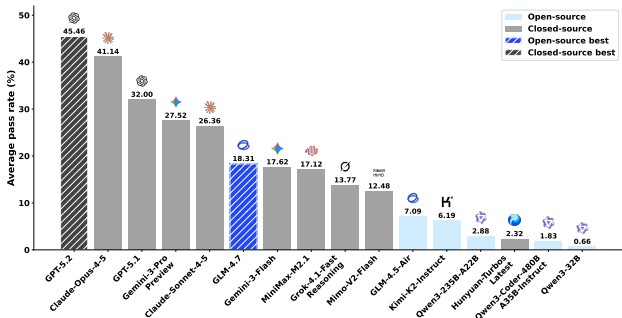

*Figure 5.* Overall model pass rate on MINIAPPBENCH

*Figure 4.* **MINIAPPEVAL vs. Previous Methods.** Unlike brittle scripts or rigid comparisons, MINIAPPEVAL integrates code inspection with dynamic execution. It complements human evaluation by verifying underlying physical principles and automating tedious testing scenarios to ensure robust assessment.

are executed via targeted JavaScript injected in the browser context for precise, deterministic control. The agent perceives rich signals (DOM, console logs, and source code; Appendix B.6.2) and selects actions (Appendix B.6.3) to probe functionality, guided by the query-specific evaluation reference $r_i$ to map requirements to verifiable checks and collect concrete evidence. The full process is recorded as a reproducible interaction trajectory (Appendix B.9).

**Scoring.** Given the customized interactivity of MINIAPPS and their grounding in real-world principles, **MINIAPPEVAL** combines static analysis with dynamic evidence to evaluate MINIAPPS along three dimensions: **Intention**, **Static**, and **Dynamic**. The evaluation reference $r_i$, which encodes expected behaviors grounded in real-world principles, guides the agent's inspection strategy but does not serve as a rigid oracle. Instead, the final judgment is based on whether the MiniApp functionally satisfies the user's request. The output is a structured score across the three dimensions, each accompanied by a detailed rationale (highlighted in red at the top of Figure 4 (b)).

MINIAPPEVAL departs from assertion-based or comparative benchmarks by directly evaluating whether a MiniApp satisfies open-ended user requirements, making it suitable for highly customized applications (the comparison shown in Figure 4).

Moreover, MINIAPPEVAL addresses key limitations of human evaluation as shown in Figure 4 (b): (i) its static analysis precisely verifies implementation logic against real-world principles; (ii) Playwright's programmatic control improves execution efficiency; and (iii) the LLM-powered evaluator leverages broad domain knowledge, often outperforming non-expert annotators on specialized tasks.

## 5 Experiments

### 5.1 Settings

All evaluations are conducted in a sandbox with deterministic seeds and fixed rendering settings. Artifacts are rendered via Playwright (headless Chromium) at multiple resolutions, including 1280×720, to test adaptive designs. Models receive identical prompts (listed in Appendix D.1) and follow a unified decoding protocol: we use officially recommended decoding parameters when available; otherwise, we apply our defaults (detailed in Appendix B.1.1). Overlong inputs are truncated, and each run is capped at 15 minutes.

Baseline models are selected to ensure breadth, currency, and reproducibility, considering: (1) **multiple model families** (Claude (Anthropic, 2025a;b), Gemini (Google DeepMind, 2025a;b), GLM (Zeng et al., 2025), GPT (OpenAI, 2025b;a), Grok (xAI, 2025), Hunyuan (Tencent Hunyuan Team et al., 2025), Kimi (Kimi Team et al., 2025), Mimo (Xiao et al., 2026), MiniMax (Chen et al., 2025), and Qwen3 (Yang et al., 2025)); (2) **a range of scales** (from lightweight to flagship); and (3) relatively **recent and representative versions** within each family.

For evaluation, we use Gemini-3-Pro-Preview as the MINIAPPEVAL driver model. It was selected from five candidate judges (Gemini-3-Pro-Preview, GPT-5.2, Claude-Sonnet-4.5, Qwen3-Coder-480B-A35B-Instruct, and GLM-4.7) because it achieved the highest agreement with expert annotations on the representative subset (Appendix B.2). Unless otherwise stated, judge decoding uses temperature = 1. Additional temperature robustness analyses are provided in Appendix B.4.

### 5.2 Main Results and Analysis

Our framework supports custom thresholds. In our experiments, we adopt $\tau = 0.8$: a MiniApp is considered successful if its minimum score across the three dimensions (Intention, Static, Dynamic) is at least this value, i.e., $\min(S^i, S^s, S^d) \geq 0.8$. This threshold is motivated by the score-band design in the evaluation prompt: 0.8–1.0 indicates strong satisfaction, 0.5–0.7 partial satisfaction, and 0.0–0.4 failure. We use the minimum across dimensions as a

*Table 2.* Performance of models on MINIAPPBENCH: Pass Rate, Token Consumption, and Inference Time

| Model | Pass Rate (%) | | | | | | | | | Avg. (%) | Tokens | Time(s) |
|---|---|---|---|---|---|---|---|---|---|---|---|---|
| | Difficulty | | | Domain | | | | | | | | |
| | Easy | Mid | Hard | Games | Science | Tools | Humanities | Viz. | Lifestyle | | | |
| *Open-Source Large Language Models* | | | | | | | | | | | | |
| Qwen3-32B | 1.59 | 0.55 | 0.00 | 0.00 | 0.57 | 0.00 | 0.00 | 2.04 | 3.70 | 0.66 | 3,470.68 | 22.16 |
| Qwen3-235B-A22B | 6.43 | 2.35 | 0.00 | 0.93 | 0.60 | 4.00 | 4.88 | 7.27 | 10.34 | 2.88 | 4,068.27 | 49.55 |
| Qwen3-Coder-480B-A35B-Instruct | 6.06 | 0.00 | 0.00 | 0.00 | 0.00 | 0.00 | 0.00 | 9.43 | 11.11 | 1.83 | 2,324.83 | 25.04 |
| Kimi-K2-Instruct | 14.17 | 5.03 | 0.00 | 3.77 | 3.11 | 4.08 | 4.88 | 17.65 | 18.52 | 6.19 | 3,435.97 | 46.76 |
| GLM-4.5-Air | 17.60 | 4.07 | 1.44 | 5.66 | 4.27 | 6.98 | 7.32 | 16.98 | 10.34 | 7.09 | 7,110.65 | 58.94 |
| GLM-4.7 | **36.30** | **15.06** | **4.41** | **12.50** | **10.49** | **20.00** | **17.07** | **35.19** | **48.39** | **18.31** | 8,936.88 | 55.58 |
| *Closed-Source Large Language Models* | | | | | | | | | | | | |
| Hunyuan-Turbos-Latest | 6.32 | 0.87 | 0.00 | 0.00 | 0.00 | 3.03 | 0.00 | 13.51 | 3.57 | 2.32 | 3,727.55 | 132.67 |
| Mimo-V2-Flash | 28.68 | 8.33 | 2.22 | 13.46 | 6.02 | 10.87 | 11.63 | 23.53 | 36.36 | 12.48 | 5,109.82 | 37.98 |
| Grok-4-1-Fast-Reasoning | 29.66 | 12.12 | 2.19 | 8.41 | 6.58 | 20.00 | 17.50 | 32.65 | 25.93 | 13.77 | 9,010.00 | 75.62 |
| MiniMax-M2.1 | 31.46 | 15.62 | 7.08 | 16.25 | 12.50 | 23.33 | 20.00 | 27.27 | 19.23 | 17.12 | 8,881.57 | 118.32 |
| Gemini-3-Flash | 32.76 | 16.89 | 4.10 | 14.95 | 10.60 | 17.95 | 18.18 | 30.61 | 41.38 | 17.62 | 6,563.28 | 50.56 |
| Gemini-3-Pro-Preview | 61.98 | 20.83 | 1.71 | 26.74 | 19.11 | 13.64 | 28.57 | 52.00 | 55.56 | 27.52 | 5,815.14 | 80.80 |
| Claude-Sonnet-4-5 | 68.22 | 14.86 | 1.79 | 16.13 | 22.30 | 29.27 | 23.81 | 47.73 | 44.83 | 26.36 | 8,586.84 | 91.43 |
| Claude-Opus-4-5 | 59.09 | 41.18 | **22.33** | 37.18 | 34.59 | 47.50 | 35.71 | 57.45 | 56.52 | 41.14 | 13,152.75 | 166.66 |
| GPT-5.1 | **74.71** | 21.37 | 3.49 | 24.14 | 18.10 | 33.33 | **45.83** | 57.78 | 64.71 | 32.00 | 11,256.15 | 154.09 |
| GPT-5.2 | 69.77 | 43.08 | 18.64 | 40.32 | 50.38 | 50.17 | 45.45 | 75.00 | 82.35 | 45.46 | 10,793.68 | 169.60 |
| **Average** | **34.05** | **13.89** | **4.34** | **14.71** | **11.64** | **18.07** | **17.55** | **31.63** | **33.30** | **17.05** | – | – |

conservative weakest-link rule, since a MiniApp should pass only when every dimension reaches the high-satisfaction band. Sensitivity analysis over thresholds 0.6, 0.7, 0.8, and 0.9 shows stable rankings (Appendix B.3). GPT-5.2 achieved the highest performance with an average pass rate of 45.46%, while the overall mean across all models was 17.05%. These results underscore the challenges current models face in generating successful MINIAPPS. The details are shown in Figure 5 and Table 2.

**Open-Source vs. Closed-Source Performance Analysis.**
Our experiments show a clear gap between open- and closed-source models, with closed-source systems consistently performing better across all difficulty levels. In contrast, benchmarks such as ArtifactsBench (Zhang et al., 2025) and WebDevJudge (Li et al., 2025a) report much smaller gaps, suggesting potential saturation or overfitting; our benchmark better avoids this issue and thus provides a more discriminative evaluation.

**Difficulty-Level Performance Analysis.** The difficulty-wise performance analysis validates the rationale behind our task difficulty gradient segmentation, showing that models with different performance levels can find their respective niches when tackling tasks of varying complexity. As shown in Table 2, the accuracy of all models decreases with increasing difficulty. Furthermore, smaller open-source models (Qwen3-32B) can handle certain tasks effectively, whereas more advanced models often struggle with more complex challenges.

**Domain-wise Performance Analysis.** As shown in Table 2, the performance varies significantly across different classes. The pass rates for the Visualization and Lifestyle categories are notably higher, exceeding 30%, with GPT-5.2 performing particularly well. This suggests that current models excel in tasks with a clear, singular objective, such as visualizations, and in tasks that just require the application of commonsense. However, for more complex categories that involve comprehensive tasks, domain-specific knowledge, and intricate engineering details, the models still exhibit some limitations.

**Model-Scale and Positioning Analysis.** Across both the Qwen and GLM families, we observe a consistent trend where increasing model scale generally leads to superior performance, validating the impact of scaling laws on complex tasks. Within the Qwen3 series, Qwen3-235B-A22B achieves a 2.88% pass rate, significantly outperforming the smaller Qwen3-32B (0.66%). This scaling trajectory is even more pronounced in the GLM series: the lightweight GLM-4.5-Air achieves a 7.09% pass rate, while the flagship GLM-4.7 reaches a substantial 18.31%, illustrating the massive performance gains derived from increased model capacity and architectural refinement.

**Performance vs. Inference Cost Analysis.** There is a strong positive correlation between performance and token consumption (0.8433), and a moderate correlation with time (0.7387), as illustrated in Figure 6, suggesting that more tokens and time generally improve performance. The correlation is measured by the Pearson correlation coefficient (Pearson, 1895). Outliers include GPT-5.2 and Gemini-3-

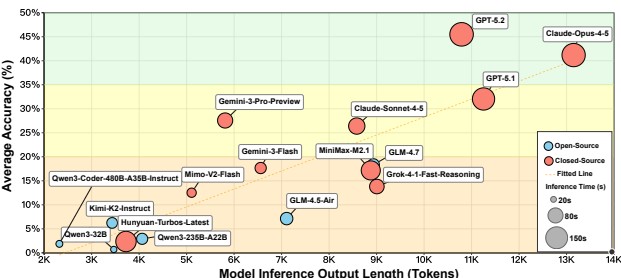

*Figure 6.* Token Length & Inference Time vs Average pass rate

*Table 3.* A**blation results (%)**. Metrics include accuracy (Acc.), precision (Prec.), recall (Rec.), and F1. The superscript arrows denote the absolute change relative to the MINIAPPEVAL.

| Exp. | Acc. | Prec. | Rec. | F1 |
|---|---|---|---|---|
| MINIAPPEVAL | **89.62** | **83.87** | **85.25** | **84.55** |
| w/o Code | 70.66 ↓18.96 | 32.73 ↓51.14 | 60.00 ↓25.25 | 42.35 ↓42.20 |
| w/o Agent | 66.48 ↓23.14 | 12.90 ↓70.97 | 53.33 ↓31.92 | 20.78 ↓63.77 |
| w/o Eval Ref | 60.12 ↓29.50 | 89.47 ↑5.60 | 46.36 ↓38.89 | 61.08 ↓23.47 |

Pro-Preview, which consume fewer tokens than models with similar performance. Hunyuan-Turbos-Latest and MiniMax-M2.1 have notably higher processing times for similar performance.

## 5.3 Ablation Study

Due to the high cost of expert annotation, ablation and human-agreement studies are conducted on a representative 200-item subset matched to the full 500-task benchmark by domain, difficulty, and model pass-rate distribution (Appendix A.2). To evaluate the impact of different components on the performance of **MINIAPPEVAL**, we conducted an ablation study on this manually labeled ground truth (GT) subset, as shown in Table 3. The full MINIAPPEVAL system (comprising *Eval-Ref*, *Code*, and *Playwright*) achieves the highest accuracy among all variants, demonstrating the overall effectiveness of the proposed evaluation framework. Removing the Eval-Ref leads to a substantial drop in recall, indicating that the Eval-Ref plays a critical role in guiding MINIAPPEVAL to attend to the correct aspects of a query and to accurately localize potential failure cases. **w/o Code** results in a sharp degradation in precision, as the judge can no longer verify implementation details (e.g., detect violations of implicit real-world principles). **w/o Agent** yields the lowest precision overall, highlighting that many interaction-dependent behaviors can only be revealed through active exploration, which are inaccessible to static inspection alone.

## 5.4 Double Blind Judge

During evaluation, we observed that for graphical queries (e.g., in the Visualization class), the agent judge could be overly lenient due to confirmation bias (Nickerson, 1998). To mitigate this, we introduce a double-blind evaluation

*Table 4.* Evaluation accuracy comparison between MINIAPPEVAL and double-blind methods.

| Model | Method | T/T | T/F | F/T | F/F | Acc. |
|---|---|---|---|---|---|---|
| Gemini-3-Pro-Pro-Preview | MINIAPPEVAL | 15 | 2 | 8 | 30 | 81.82 |
| | Double-Blind | 11 | 6 | 2 | 36 | 85.45 ↑3.63 |
| GPT-5.2 | MINIAPPEVAL | 16 | 3 | 8 | 28 | 80.00 |
| | Double-Blind | 12 | 7 | 2 | 34 | 83.63 ↑3.63 |
| Claude-Opus-4.5 | MINIAPPEVAL | 17 | 3 | 9 | 26 | 78.18 |
| | Double-Blind | 11 | 9 | 0 | 35 | 83.63 ↑5.45 |

procedure (detailed in Appendix C): the judge first evaluates the output without seeing the query, and then checks it against the user requirements for the final decision. We apply this protocol to 55 graphical queries. As shown in Table 4, it improves accuracy and better identifies negative samples, supporting our hypothesis and offering a more reliable setup for purely visual tasks.

## 5.5 Validation of Evaluation Effectiveness

To validate the effectiveness and reliability of **MINIAPPE-VAL**, we conducted a human agreement study with four experts on 200 items from each of three representative models spanning different performance tiers: **low-** (GLM-4.7), **mid-** (Gemini-3-pro-preview), and **high-performing** (GPT-5.2) (600 outputs total); each output was annotated by all four experts (2,400 annotations).

We first assessed inter-rater reliability using Fleiss' Kappa (Fleiss, 1971), obtaining $\kappa = 0.89$. Using the aggregated expert labels as reference, we then computed MINIAPPEVAL–human agreement across the three models to cover different quality regimes. As shown in Table 5, MINIAPPEVAL achieves strong agreement with humans on the 200-item subset, with an average F1 of 92.4%. Additional repeated-generation and repeated-evaluation analyses are provided in Appendix B.5.

*Table 5.* MINIAPPEVAL–human agreement across models with different performance levels on the 200-item subset. JPass denotes the judge pass rate.

| Model | JPass (%) | Acc. (%) | F1 (%) |
|---|---|---|---|
| GLM-4.7 | 18.5 | 94.5 | 94.5 |
| Gemini-3-Pro-Preview | 28.0 | 90.7 | 90.7 |
| GPT-5.2 | 46.0 | 92.3 | 92.4 |

## 6 Conclusion

In conclusion, we introduce **MINIAPPBENCH**, the first benchmark for evaluating principle-driven interactive application generation, addressing key gaps left by prior benchmarks. We further propose MINIAPPEVAL, an agentic, browser-based evaluation framework that enables comprehensive and automated assessment of MINIAPPS. Our experiments show that current LLMs still struggle to generate high-quality MINIAPPS, while MINIAPPEVAL aligns closely with human judgments, providing a reliable method for future research.

## Acknowledgements

This work was supported by Ant Group Research Intern Program.

## Impact Statement

This work advances the field of Machine Learning by providing a framework that could facilitate a transition in human-AI interaction from static text responses to the collaborative creation of dynamic, interactive applications. By introducing MINIAPPBENCH, we support the potential democratization of software creation, empowering non-experts to act as architects of experience who could translate abstract intents into functional, executable artifacts. In educational settings, this paradigm shift may foster greater equity by enabling the generation of personalized, interactive pedagogical tools. While this open-ended medium presents new reliability challenges, our work offers a systematic way to identify failure modes, ultimately guiding the development of more robust, transparent, and socially responsible AI systems.

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

# A  Data Construction

## A.1  Domain Classification

The two-level taxonomy of queries is constructed in two stages. First, we employ large language models (LLMs) to categorize queries collected from real-world users, producing an initial taxonomy. Human experts then review and refine the taxonomy to develop a more logical and coherent classification scheme. The final taxonomy consists of six coarse-grained domains: Science, Games, Tools, Humanities, Lifestyle, and Visualization. Each domain is further divided into finer-grained subclasses, as outlined in Table 6.

To ensure a comprehensive evaluation of model capabilities, we carefully considered category distribution during dataset construction. While the dataset was initialized based on real-world user query distributions, we applied controlled rebalancing to avoid excessive concentration in highly frequent categories. For instance, although game-related queries are highly prevalent in online user traffic, their proportion was moderately reduced while remaining sufficiently represented in the final dataset.

*Table 6.* The Data Domain Classification

| Domain | Subclass | Count | Ratio (%) |
|---|---|---|---|
| Science | Chemical | 46 | 9.20 |
| | Biological Systems | 44 | 8.80 |
| | Physics | 37 | 7.40 |
| | Virtual Laboratory | 35 | 7.00 |
| | Geometry | 25 | 5.00 |
| | **Total (Science)** | **187** | **37.40** |
| Games | Logic | 28 | 5.60 |
| | Projectile | 25 | 5.00 |
| | Reflex | 16 | 3.20 |
| | Edutainment | 16 | 3.20 |
| | Systemic Simulation | 15 | 3.00 |
| | Casual | 11 | 2.20 |
| | Card | 10 | 2.00 |
| | **Total (Games)** | **121** | **24.20** |
| Tools | Schedule | 21 | 4.20 |
| | Creative Tools | 18 | 3.60 |
| | Computational Tools | 15 | 3.00 |
| | Data Lookup | 3 | 0.60 |
| | **Total (Tools)** | **57** | **11.40** |
| Humanities | Skill Acquisition | 26 | 5.20 |
| | Concept Deconstruction | 13 | 2.60 |
| | Culture | 8 | 1.60 |
| | **Total (Humanities)** | **47** | **9.40** |
| Lifestyle | Health | 14 | 2.80 |
| | Toys | 10 | 2.00 |
| | Roleplay | 8 | 1.60 |
| | **Total (Lifestyle)** | **32** | **6.40** |
| Visualization | SVG | 25 | 5.00 |
| | Statical | 23 | 4.60 |
| | Art | 8 | 1.60 |
| | **Total (Visualization)** | **64** | **11.20** |
| | **Grand Total** | **500** | **100.00** |

*Table 7.* Representativeness of the 200-item subset compared with the full 500-task benchmark.

| Domain Distribution | | | Difficulty Distribution | | | Model Pass Rate | | | |
|---|---|---|---|---|---|---|---|---|---|
| Domain | 200 (%) | 500 (%) | Level | 200 (%) | 500 (%) | Model | 500 (%) | 200 (%) | Δ |
| Science | 35.0 | 37.4 | Easy | 28.5 | 29.4 | GPT-5.2 | 45.46 | 46.00 | +0.54 |
| Games | 23.0 | 24.2 | Medium | 42.0 | 39.2 | Claude-Opus-4-5 | 41.14 | 41.50 | +0.36 |
| Tools | 13.0 | 11.4 | Hard | 29.5 | 31.4 | Gemini-3-Pro-Preview | 27.52 | 28.00 | +0.48 |
| Humanities | 8.0 | 9.4 | | | | GLM-4.7 | 18.31 | 18.50 | +0.19 |
| Lifestyle | 7.5 | 6.4 | | | | Qwen3-32B | 0.66 | 0.50 | -0.16 |
| Visualization | 13.5 | 11.2 | | | | | | | |

*Table 8.* Domain by difficulty distribution of the 500-task benchmark.

| Domain | Easy | Medium | Hard | Total |
|---|---|---|---|---|
| Science | 46 (25%) | 72 (38%) | 69 (37%) | 187 |
| Games | 23 (19%) | 49 (40%) | 49 (40%) | 121 |
| Tools | 13 (23%) | 27 (47%) | 17 (30%) | 57 |
| Visualization | 30 (54%) | 20 (36%) | 6 (11%) | 56 |
| Humanities | 16 (34%) | 18 (38%) | 13 (28%) | 47 |
| Lifestyle | 19 (59%) | 10 (31%) | 3 (9%) | 32 |
| **Total** | **147 (29%)** | **196 (39%)** | **157 (31%)** | **500** |

## A.2 Representativeness of the Expert-Annotated Subset

We further verify that the expert-annotated 200-item subset is representative of the full 500-task benchmark. The subset closely matches the full benchmark in terms of domain distribution, difficulty distribution, and pass rates across representative models. The detailed statistics are summarized in Table 7.

## A.3 Dataset Composition and Augmentation

The final benchmark contains 77 direct real-user queries (15.4%) and 423 LLM-augmented variants (84.6%). Augmentation is used to counter the simplicity and domain skew of raw online queries while preserving each seed's topic and intent.

**Example.** Seed: "Simulate the solar system." Augmented: "Create a solar system planetary simulator showing a planet on a flattened elliptical orbit. Allow users to shade the sector area swept over a time interval, verifying that equal time intervals sweep equal areas. Display the planet's instantaneous velocity vector in real time."

The dataset follows a balanced difficulty distribution of 30% Easy, 40% Medium, and 30% Hard, the details are shown in 8.

## A.4 Screening Guidelines

### A.4.1 CUSTOMIZED INTERACTION

To ensure that MINIAPPBENCH targets *query-specific interaction* rather than conventional CRUD-oriented web applications, we screen candidate queries by examining whether the requested behavior requires synthesizing interaction logic that cannot be reduced to standard CRUD workflows.

Concretely, a query is labeled as requiring customized interaction if it satisfies **at least one** of the following criteria. These criteria cover interaction mechanics, runtime behavior, and user exploration patterns.

✔ **Multi-step state transitions.** The task requires maintaining and updating non-trivial internal states across multiple user actions (e.g., "simulate one week of choices", "step-by-step experiment", "undo/redo", "scenario branching"), beyond simple record manipulation.

✔ **Custom interaction operators.** The task involves interaction primitives that do not commonly appear in standard CRUD interfaces, such as dragging, drawing, manipulating sliders to control simulations, playing a game, interactive diagram exploration, timeline scrubbing, or parameter sweeping.

✔ **Dynamic rules grounded in the query.** The runtime behavior must obey explicitly specified or strongly implied rules unique to the query, such as physical laws (gravity, conservation), temporal constraints (a week has seven days), geometric constraints, scoring rules in games, or procedural generation rules.

✔ **Open-ended user exploration.** The interface is intended to support iterative exploration of a concept (e.g., "interactive

visualization to understand ...", "what-if analysis"), where value arises from interaction rather than static content presentation.

✔ **Non-trivial edge-case handling.** The query implies boundary conditions that affect interaction logic (e.g., invalid parameter ranges, impossible states, constraint violations), requiring tailored runtime checks beyond standard input validation.

We **exclude** queries that can be effectively solved by: (i) static information presentation (e.g., "show me an introduction to ..."), or (ii) standard CRUD-style applications (e.g., "create a webpage to add/edit/delete notes"), where the interaction can be implemented with a generic form-list template and does not require query-specific dynamics.

During screening, each candidate query is independently assessed by two annotators according to the above criteria. Disagreements are resolved through discussion, and borderline cases are retained only when the interaction behavior is primarily determined by query-specific rules rather than templated CRUD patterns.

### A.4.2 REAL-WORLD PRINCIPLE

In addition to customized interaction, we require each query to involve at least one **real-world principle** that constrains the MiniApp's behavior. Here, a principle refers to an implicit or explicit rule about how the world should work (e.g., physical laws, temporal constraints, domain conventions, or commonsense invariants) that must be **operationalized** in an executable artifact.

**Principle taxonomy.** Our principle categorization follows the European Research Area (ERA), covering four broad areas: *Life Sciences*, *Physical Sciences and Engineering*, *Social Sciences and Humanities*, and *Health and Medicine*. Each query is annotated with the area(s) of principle it primarily relies on (e.g., conservation laws in a physics simulation; biological processes in a cell-cycle demo; historical timelines and causal narratives in humanities; dosage/health constraints in medicine).

**HTML-expressibility requirement.** Crucially, we only retain queries whose underlying principles can be faithfully expressed and verified through a browser executable interface. Our screening assumes the following executable web decomposition: **HTML** represents world states and structural relationships, **CSS** determines perceptual salience, and **JavaScript** encodes causal dependencies, temporal evolution, and interaction logic, together forming an executable world model. Therefore, a query passes the principle screening only if the principle can be mapped to at least one of the following HTML expressible forms:

✔ **State representation:** the relevant entities, attributes, and constraints can be represented as DOM elements and state variables (e.g., positions, counts, schedules, scores).

✔ **Rule execution:** the principle can be implemented as deterministic or stochastic update rules in JavaScript that govern state transitions over time and user interactions (e.g., numerical integration for motion, discrete event simulation, rule-based scoring).

✔ **Perceptual grounding:** the principle's outcomes can be rendered and inspected via visual encodings or UI feedback (e.g., trajectories, charts, alerts, invariants displayed as diagnostics).

We **exclude** queries whose required principles are not meaningfully capturable in an offline, self-contained browser setting, such as tasks requiring external sensors, proprietary databases, real-time web access, or unverifiable claims that cannot be grounded in executable state-transition logic. This ensures that every retained query admits a MiniApp implementation where principle adherence is both implementable and testable within HTML/CSS/JavaScript.

### A.5 Data Format

The evaluation dataset is stored as a JSON array. Each element corresponds to one MiniApp specification and its LLM-generated evaluation reference. Each record contains six fields: `index`, `class`, `subclass`, `query`, `level`, and `eval-reference` (a JSON-serialized string).



**Data Format**

```
{
  "index": 137,
  "class": "Tools",
  "subclass": "Creative Tools",
  "query": "Design a timeline visualization editor that renders a horizontal timeline
      on a canvas, allowing users to add nodes, drag to reposition them, set colors and
      labels, and supports zooming and exporting as an image.",
  "level": "Hard",
  "eval-reference": "{ \"intention\": [...], \"static\": [...], \"dynamic\": [...] }"
}
```

</div>

**Fields.** `index` is a unique identifier (1-based) within the file. `class` and `subclass` denote the coarse- and fine-grained categories. `query` is the natural-language specification used for generation. `level` is the difficulty tag (`Easy`/`Mid`/`Hard`). `eval-reference` encodes the evaluation reference in three dimensions (`intention`, `static`, `dynamic`) and is parsed by the evaluator when needed.

# B  MINIAPPEVAL

## B.1  Settings

### B.1.1  MODELS' DECODING PROTOCOL

We follow each model's official API documentation or default demo settings when available. For models without explicit recommendations, we adopt commonly used default values to ensure fair comparison. The specific settings shown in Table 9.

*Table 9.* Decoding settings for all evaluated models.

| Model | Temperature | Top-$p$ | Max tokens |
|---|---|---|---|
| GPT-5.2 | 1.0 | 1.0 | 128,000 |
| GPT-5.1 | 1.0 | 1.0 | 400,000 |
| Claude-Opus-4.5 | 1.0 | 1.0 | 200,000 |
| Claude-Sonnet-4.5 | 1.0 | 1.0 | 200,000 |
| Gemini-3-Pro-Preview | 0.8 | 0.95 | 65,536 |
| Gemini-3-Flash | 0.8 | 0.95 | 65,536 |
| GLM-4.7 | 1.0 | 0.95 | 131,072 |
| GLM-4.5-Air | 1.0 | 0.95 | 96,000 |
| MiniMax-M2.1 | 1.0 | 1.0 | 204,800 |
| Grok-4.1-Fast-Reasoning | 1.0 | 1.0 | 30,000 |
| Mimo-V2-Flash | 1.0 | 1.0 | 32,768 |
| Kimi-K2-Instruct | 1.0 | 1.0 | 256,000 |
| Qwen3-235B-A22B | 1.0 | 1.0 | 38,912 |
| Qwen3-Coder-480B-A35B-Instruct | 1.0 | 1.0 | 65,536 |
| Qwen3-32B | 1.0 | 1.0 | 32,768 |
| Hunyuan-Turbos-Latest | 1.0 | 1.0 | 256,000 |

### B.1.2  TWO MINIAPPS GENERATION FORMATS

To more comprehensively evaluate model capabilities while reducing interference from output formatting, our evaluation supports two generation modes: (1) a single-file HTML mode and (2) a React framework mode. For both modes, the pipeline automatically extracts the generated code, builds a runnable project, launches it in a sandboxed environment, and then completes the evaluation. The recommended file structure for the React mode is shown below. Prompts for both generation formats are provided in D.1.

**The Recommended File Structure for the React Mode**

```
template/
|
|-- src/
| |-- App.tsx # Main page component (business code goes here)
| |-- main.tsx # React entry point, mounted to #root
| |-- index.css # Global styles (plain CSS)
| |-- base.js
| \-- global.d.ts # Global type definitions (provide basic declarations)
|
|-- index.html # HTML entry point, containing <div id="root">
|-- package.json # Dependencies + scripts: dev/build/preview
| # Note: If postcss.config.js is used, must include "autoprefixer" and "postcss" in
    devDependencies
|-- vite.config.ts # Vite configuration (React plugin + base.js entry)
|-- tsconfig.json # TypeScript configuration (if using project references, must
    include references)
|-- tsconfig.node.json # TypeScript Node configuration (for vite.config.ts, must be
    generated if tsconfig.json has references)
\-- postcss.config.js # PostCSS configuration (if using autoprefixer, package.json
    must include autoprefixer and postcss dependencies)

Important Notes:
1. If autoprefixer is used in postcss.config.js, package.json's devDependencies must
    include:
  "autoprefixer": "^10.4.14",
  "postcss": "^8.4.31"
2. If tsconfig.json uses the "references" field to reference tsconfig.node.json, then
    tsconfig.node.json must be generated. Example content:
  {{
   "compilerOptions": {{
    "composite": true,
    "skipLibCheck": true,
    "module": "ESNext",
    "moduleResolution": "bundler",
    "allowSyntheticDefaultImports": true,
    "strict": true
   }},
   "include": ["vite.config.ts"]
  }}
```

### B.1.3 POSITIVE/NEGATIVE LABELING

We convert the three-dimensional evaluation scores into a binary label. A MiniApp is marked as **positive** if all three dimension scores are greater than or equal to a predefined threshold:

$$\min(s_{\text{intention}}, s_{\text{static}}, s_{\text{dynamic}}) \geq \tau, \tag{3}$$

where $\tau = 0.8$ in the main setting. Otherwise, it is labeled as **negative**.

### B.2 Judge Model Selection

We compare five candidate judge models on the 200-item expert-annotated subset: GPT-5.2, Claude-Sonnet-4.5, GLM-4.7, Qwen3-Coder-480B-A35B-Instruct, and Gemini-3-Pro-Preview. Tables 10, 11, 12, 13, and 14 report agreement with expert labels across thresholds and representative generation models. Gemini-3-Pro-Preview achieves the highest average F1 (92.4%) among the five candidate judges, so we use it as the MINIAPPEVAL driver model.

### B.3 Threshold Selection

We select $T = 0.8$ because it matches the high-satisfaction score band in the evaluation prompt: 0.8–1.0 indicates strong satisfaction, 0.5–0.7 partial satisfaction, and 0.0–0.4 failure. Under the minimum-score rule and the selected Gemini-3-Pro-Preview judge, $T = 0.8$ achieves the best average F1 among tested thresholds (Table 14). Across thresholds 0.6, 0.7, 0.8, and 0.9, model rankings remain highly stable (Tables 15 and 16), supporting the robustness of this choice. Relative to

*Table 10.* GPT-5.2 judge: 200-sample human-agreement results across thresholds and representative generation models.

| $T$ | GPT-5.2 | | | GLM-4.7 | | | Gemini-3-Pro-Preview | | |
|---|---|---|---|---|---|---|---|---|---|
| | JPass | Acc. | F1 | JPass | Acc. | F1 | JPass | Acc. | F1 |
| 0.6 | 35.5 | 73.0 | 65.8 | 28.0 | 78.5 | 53.8 | 30.5 | 78.0 | 61.4 |
| 0.7 | 25.0 | 73.5 | 61.3 | 16.5 | 91.0 | 74.3 | 8.5 | 76.0 | 31.4 |
| 0.8 | 11.0 | 67.5 | 40.4 | 4.5 | 83.0 | 26.1 | 7.0 | 75.5 | 26.9 |
| 0.9 | 0.0 | 56.5 | – | 0.0 | 81.5 | – | 0.0 | 73.5 | – |

*Table 11.* Claude-Sonnet-4.5 judge: 200-sample human-agreement results across thresholds and representative generation models.

| $T$ | GPT-5.2 | | | GLM-4.7 | | | Gemini-3-Pro-Preview | | |
|---|---|---|---|---|---|---|---|---|---|
| | JPass | Acc. | F1 | JPass | Acc. | F1 | JPass | Acc. | F1 |
| 0.6 | 53.5 | 70.0 | 69.1 | 43.5 | 78.0 | 68.6 | 48.5 | 70.0 | 55.2 |
| 0.7 | 42.5 | 82.0 | 79.1 | 20.0 | 76.5 | 49.5 | 36.5 | 82.0 | 67.3 |
| 0.8 | 28.5 | 85.0 | 79.2 | 9.0 | 76.5 | 33.8 | 21.5 | 91.0 | 77.5 |
| 0.9 | 23.5 | 80.0 | 70.1 | 2.0 | 74.5 | 10.5 | 6.5 | 87.0 | 48.0 |

$T = 0.8$, Spearman's $\rho$ remains above 0.97 and Kendall's $\tau$ is no lower than 0.90.

We also compare alternative aggregation criteria. Average-score and median-score rules yield lower optimal F1 than the minimum-score rule (Tables 17 and 18), supporting the conservative weakest-link design: a MiniApp should pass only when every evaluation dimension reaches the high-satisfaction band.

### B.4 Temperature Robustness
We evaluate judge temperatures 0.5, 0.8, 1.0, and 1.2 under threshold 0.8. The results show limited sensitivity, and temperature $= 1$ achieves the best or near-best F1 across representative generation models (Table 19).

### B.5 Evaluation Stability
We assess stability from two perspectives: multiple generations from the same model and repeated evaluations of the same sample. Across five runs, pass-rate standard deviations remain low, and per-query agreement is high, with Fleiss' $\kappa > 0.90$ in all settings. Details are shown in Tables 20–23

### B.6 Environment Setup
Our evaluation framework conducts agent assessments in web-based environments, enabling comprehensive evaluation of GUI agents through automated browser interaction and code analysis. The evaluation system is implemented through a standardized evaluation script that provides a consistent interface for assessing agent-generated web applications. In the following sections, we detail the environment design for web-based agent evaluation.

#### B.6.1 ENVIRONMENT INFRASTRUCTURE
We design an interactive web-based evaluation environment using browser automation technology. The environment leverages Playwright as the browser automation platform through the Model Context Protocol (MCP) server interface, enabling high compatibility with real-world web applications while maintaining full control over the execution environment. This setup allows us to simulate user interactions such as mouse clicks, keyboard input, and form submissions, which are essential for evaluating GUI agents' capabilities. The browser automation framework supports real-time observation and logging of DOM states, facilitating fine-grained analysis and reproducibility of agent behavior. All evaluation episodes are initialized from a clean browser state to ensure consistent starting conditions for each evaluation episode. The evaluation system supports two complementary modes: standard mode, where agents interact with live web applications via URLs with full browser automation capabilities, and code-only mode, where evaluation is performed solely based on HTML and JavaScript code analysis without browser access. This dual-mode design enables flexible evaluation strategies, allowing assessment of both runtime behavior and static code quality.

#### B.6.2 OBSERVATION SPACE
In our evaluation framework, the observation space is designed to ensure comprehensive evaluation of web-based GUI agents by capturing both structural and semantic aspects of web pages. It comprises two complementary modalities: DOM structure snapshots and source code access. The DOM snapshot is obtained through the Playwright MCP server's `browser_evaluate` interface, which provides a complete representation of the page's hierarchical structure, including

*Table 12.* GLM-4.7 judge: 200-sample human-agreement results across thresholds and representative generation models.

| $T$ | GPT-5.2 | | | GLM-4.7 | | | Gemini-3-Pro-Preview | | |
|-----|-------|------|------|-------|------|------|-------|------|------|
| | JPass | Acc. | F1 | JPass | Acc. | F1 | JPass | Acc. | F1 |
| 0.6 | 77.0 | 66.5 | 72.2 | 48.0 | 66.5 | 49.6 | 67.0 | 59.5 | 56.7 |
| 0.7 | 66.0 | 77.5 | 79.5 | 41.0 | 72.5 | 53.8 | 53.0 | 73.5 | 66.7 |
| 0.8 | 58.0 | 85.5 | 85.7 | 29.0 | 72.5 | 42.1 | 32.0 | 75.5 | 58.1 |
| 0.9 | 28.0 | 74.5 | 64.3 | 10.0 | 79.5 | 28.1 | 19.5 | 77.0 | 50.0 |

*Table 13.* Qwen3-Coder-480B-A35B-Instruct judge: 200-sample human-agreement results across thresholds and representative generation models.

| $T$ | GPT-5.2 | | | GLM-4.7 | | | Gemini-3-Pro-Preview | | |
|-----|-------|------|------|-------|------|------|-------|------|------|
| | JPass | Acc. | F1 | JPass | Acc. | F1 | JPass | Acc. | F1 |
| 0.6 | 85.5 | 58.0 | 67.4 | 55.5 | 63.0 | 50.0 | 62.0 | 64.5 | 59.9 |
| 0.7 | 81.5 | 62.0 | 69.6 | 50.5 | 68.0 | 53.6 | 55.0 | 71.5 | 65.0 |
| 0.8 | 75.5 | 68.0 | 73.1 | 39.0 | 79.5 | 64.3 | 50.5 | 76.0 | 68.8 |
| 0.9 | 59.5 | 84.0 | 84.5 | 21.5 | 83.0 | 57.5 | 31.5 | 67.0 | 43.1 |

all HTML elements, their attributes, text content, and accessibility information. This structural information enables agents to understand the page layout, identify interactive elements, and navigate the interface effectively. Additionally, when available, agents can access the HTML and JavaScript source code directly, which provides insights into the implementation details, event handlers, and application logic. This dual-modality approach reflects the varying capabilities of different agent architectures. For example, agents that have been specifically trained on web environments often possess strong grounding abilities and can rely on DOM snapshots alone. In contrast, general-purpose language models typically benefit significantly from the semantic and structural information provided by both DOM structure and source code. By supporting both modalities, our framework enables fair and informative evaluation across a wide range of agents, ensuring robust assessment under diverse web application contexts and UI layouts. Notably, the framework explicitly prohibits the use of visual screenshots or rendering-based analysis, focusing exclusively on structural and semantic information to ensure objective and reproducible evaluation.

### B.6.3 ACTION SPACE

In our evaluation framework, the action space consists of core types of user interactions that an agent can perform to interact with web applications. These actions, summarized in Table 24, enable the agent to effectively interact with graphical user interfaces across a wide range of web applications.

This comprehensive action space allows agents to perform complex multi-step interactions, test dynamic behaviors, and verify application functionality across diverse web application scenarios. The combination of basic interaction actions (`browser_click`, `browser_type`, `browser_fill_form`) with advanced programmatic capabilities (`browser_evaluate`, `browser_wait_for`) enables thorough evaluation of both static UI elements and dynamic interactive behaviors. The framework emphasizes that all interactions must be verified through actual DOM state changes rather than assumptions, ensuring that evaluation results reflect genuine application capabilities rather than inferred behavior.

### B.7 The Pipeline of Agentic Evaluation

We design a one-click evaluation pipeline. Given only an OpenAI-compatible API endpoint, the system automatically runs the entire workflow, including loading queries, generating MINIAPPS, and evaluating the generated artifacts, while recording detailed logs. It supports multiple modes: generation can be performed in either HTML mode or React mode; evaluation includes, but is not limited to, MINIAPPEVAL, evaluation without code access, and evaluation without evaluation references. The pipeline also supports batched execution, substantially reducing evaluation overhead. Moreover, by standardizing both the generation scaffold and the evaluation environment, it minimizes external confounding factors and improves the fairness of experimental results. The pseudo-code of the workflow is shown below.

*Table 14.* Gemini-3-Pro-Preview judge: 200-sample human-agreement results across thresholds and representative generation models.

| $T$ | GPT-5.2 | | | GLM-4.7 | | | Gemini-3-Pro-Preview | | |
|---|---|---|---|---|---|---|---|---|---|
| | JPass | Acc. | F1 | JPass | Acc. | F1 | JPass | Acc. | F1 |
| 0.6 | 57.5 | 85.6 | 89.1 | 38.5 | 77.3 | 65.3 | 45.5 | 89.8 | 90.0 |
| 0.7 | 55.0 | 88.2 | 90.6 | 36.0 | 81.8 | 68.2 | 41.5 | 80.2 | 78.0 |
| 0.8 | 46.0 | 92.3 | 92.4 | 18.5 | 94.5 | 94.5 | 28.0 | 90.7 | 90.7 |
| 0.9 | 32.0 | 68.6 | 68.8 | 11.5 | 79.9 | 52.3 | 23.0 | 58.4 | 36.9 |

*Table 15.* Rankings across different thresholds. Pass rates are shown with ranks in parentheses.

| Model | $T = 0.6$ | $T = 0.7$ | $T = 0.8$ | $T = 0.9$ |
|---|---|---|---|---|
| GPT-5.2 | 71.1 (1) | 62.0 (1) | 45.5 (1) | 35.4 (1) |
| Claude-Opus-4.5 | 62.0 (2) | 52.6 (2) | 41.1 (2) | 23.1 (2) |
| GPT-5.1 | 55.2 (3) | 45.5 (3) | 32.0 (3) | 21.1 (3) |
| Gemini-3-Pro-Preview | 47.1 (4) | 36.8 (4) | 27.5 (4) | 13.8 (5) |
| Claude-Sonnet-4.5 | 42.4 (5) | 33.6 (5) | 26.4 (5) | 14.2 (4) |
| GLM-4.7 | 35.5 (7) | 26.8 (8) | 18.3 (6) | 9.3 (7) |
| Gemini-3-Flash | 41.5 (6) | 29.8 (6) | 17.6 (7) | 8.5 (8) |
| MiniMax-M2.1 | 34.3 (8) | 27.2 (7) | 17.2 (8) | 10.2 (6) |
| Grok-4.1-Fast-Reasoning | 29.4 (9) | 22.4 (9) | 13.7 (9) | 6.6 (9) |
| Mimo-V2-Flash | 27.9 (10) | 18.9 (10) | 12.5 (10) | 6.2 (10) |
| GLM-4.5-Air | 16.9 (11) | 11.3 (11) | 7.1 (11) | 2.4 (12) |
| Kimi-K2-Instruct | 13.3 (12) | 8.9 (12) | 6.2 (12) | 2.5 (11) |
| Qwen3-235B-A22B | 8.4 (13) | 4.4 (13) | 2.9 (13) | 1.1 (15) |
| Hunyuan-Turbos-Latest | 9.4 (14) | 5.3 (15) | 2.3 (14) | 1.7 (13) |
| Qwen3-Coder-480B-A35B-Instruct | 7.3 (15) | 4.8 (14) | 1.8 (15) | 1.4 (14) |
| Qwen3-32B | 3.5 (16) | 2.4 (16) | 0.7 (16) | 0.2 (16) |

*Table 16.* Ranking stability relative to $T = 0.8$.

| Comparison | Spearman's $\rho$ | Kendall's $\tau$ | Max Rank Diff. |
|---|---|---|---|
| $T = 0.6$ vs. $T = 0.8$ | 0.9941 | 0.9667 | 1 |
| $T = 0.7$ vs. $T = 0.8$ | 0.9824 | 0.9333 | 2 |
| $T = 0.9$ vs. $T = 0.8$ | 0.9765 | 0.9000 | 2 |

*Table 17.* Average-score criterion: 200-sample human-agreement results across thresholds.

| $T$ | GLM-4.7 | | GPT-5.2 | | Gemini-3-Pro-Preview | |
|---|---|---|---|---|---|---|
| | Acc. | F1 | Acc. | F1 | Acc. | F1 |
| 0.6 | 52.6 | 54.1 | 69.3 | 80.0 | 62.9 | 74.0 |
| 0.7 | 61.7 | 56.3 | 72.5 | 81.6 | 72.6 | 79.1 |
| 0.8 | 73.4 | 63.7 | 81.7 | 86.8 | 79.2 | 81.3 |
| 0.9 | 81.8 | 65.9 | 84.3 | 87.1 | 75.1 | 70.3 |

*Table 18.* Median-score criterion: 200-sample human-agreement results across thresholds.

| $T$ | GLM-4.7 | | GPT-5.2 | | Gemini-3-Pro-Preview | |
|---|---|---|---|---|---|---|
| | Acc. | F1 | Acc. | F1 | Acc. | F1 |
| 0.6 | 57.1 | 54.8 | 72.5 | 81.6 | 69.5 | 77.6 |
| 0.7 | 59.1 | 54.0 | 76.5 | 83.8 | 76.1 | 80.5 |
| 0.8 | 69.5 | 61.2 | 81.0 | 86.4 | 76.1 | 78.7 |
| 0.9 | 76.0 | 61.9 | 79.7 | 83.9 | 69.5 | 66.7 |

*Table 19.* Temperature sensitivity of Gemini-3-Pro-Preview judge at threshold 0.8.

| Temp. | GPT-5.2 | | | GLM-4.7 | | | Gemini-3-Pro-Preview | | |
|---|---|---|---|---|---|---|---|---|---|
| | JPass | Acc. | F1 | JPass | Acc. | F1 | JPass | Acc. | F1 |
| 0.5 | 54.5 | 89.0 | 88.8 | 20.5 | 97.0 | 92.3 | 35.5 | 91.0 | 85.5 |
| 0.8 | 52.5 | 91.0 | 90.6 | 18.5 | 98.0 | 94.6 | 33.5 | 93.0 | 88.3 |
| 1.0 | 46.0 | 92.3 | 92.4 | 18.5 | 94.5 | 94.5 | 28.0 | 90.7 | 90.7 |
| 1.2 | 45.0 | 93.5 | 92.7 | 15.5 | 97.0 | 91.2 | 26.5 | 95.0 | 90.6 |

*Table 20.* Pass-rate stability over five generations from the same model.

| Generation Model | Run 1 | Run 2 | Run 3 | Run 4 | Run 5 | Std. | CV |
|---|---|---|---|---|---|---|---|
| GPT-5.2 | 46.0 | 48.5 | 46.0 | 49.0 | 49.0 | 1.58 | 0.0330 |
| GLM-4.7 | 18.5 | 17.5 | 18.5 | 16.5 | 17.5 | 0.84 | 0.0474 |
| Gemini-3-Pro-Preview | 28.0 | 30.0 | 28.5 | 30.5 | 28.5 | 1.14 | 0.0393 |

*Table 21.* Per-query consistency over five generations.

| Generation Model | 5/5 Consensus | ≥4/5 Consensus | Fleiss' $\kappa$ |
|---|---|---|---|
| GPT-5.2 | 93.5 | 100.0 | 0.948 |
| GLM-4.7 | 94.5 | 99.5 | 0.924 |
| Gemini-3-Pro-Preview | 89.5 | 100.0 | 0.905 |

*Table 22.* Pass-rate stability over five repeated evaluations of the same samples.

| Generation Model | Eval 1 | Eval 2 | Eval 3 | Eval 4 | Eval 5 | Std. | CV |
|---|---|---|---|---|---|---|---|
| GPT-5.2 | 46.0 | 47.0 | 47.5 | 44.5 | 46.5 | 1.14 | 0.025 |
| GLM-4.7 | 18.5 | 18.5 | 18.0 | 18.0 | 17.5 | 0.42 | 0.023 |
| Gemini-3-Pro-Preview | 28.0 | 29.0 | 27.5 | 28.5 | 30.0 | 1.00 | 0.035 |

*Table 23.* Per-query consistency over five repeated evaluations.

| Generation Model | 5/5 Consensus | ≥4/5 Consensus | Fleiss' $\kappa$ |
|---|---|---|---|
| GPT-5.2 | 95.0 | 100.0 | 0.960 |
| GLM-4.7 | 96.5 | 100.0 | 0.953 |
| Gemini-3-Pro-Preview | 95.5 | 100.0 | 0.959 |

*Table 24.* Summary of action types in the web-based evaluation environment.

| Action | Description |
|---|---|
| **browser_click** | Simulates mouse clicks on UI control elements. Supports configurable mouse buttons (left, right, middle) and both single and double clicks. Commonly used for selecting items, activating controls, or triggering events. |
| **browser_type** | Simulates keyboard input for entering text, pressing keys, or invoking shortcuts (e.g., Ctrl+C, Enter). Enables fine-grained control over application behavior and supports both functional input and text entry. |
| **browser_fill_form** | Fills form fields with specified values, supporting various input types including text inputs, checkboxes, radio buttons, and dropdown selections. Allows batch form filling for efficient interaction with complex forms. |
| **browser_evaluate** | Executes JavaScript code to query DOM state or perform complex operations. Enables agents to extract information, manipulate page elements, or verify application state programmatically. Particularly useful for analyzing CSS styles, color schemes, and dynamic content. |
| **browser_wait_for** | Waits for specific conditions such as element appearance, text changes, or custom JavaScript predicates. Essential for handling asynchronous operations and ensuring elements are ready before interaction. |

*Algorithm 1.* QuickStart: Generate Project → Build Artifacts → Launch/Prepare Page → (Optional) Auto-Evaluation & Aggregation

**Require:** Query file $Q$ (JSON with `query/reference`), index set $\mathcal{I}$, output root $O$, raw output root $R$, evaluation platform dir $P$, AWorld dir $W$, options, port, timeouts, model config (optional)

**Ensure:** Generated page/project dirs, evaluation logs & result files, optional `time_token` summary

1: **Procedure** QUICKSTART($args$)
2: $Q \leftarrow$ RESOLVEPATH($args$.`csv_file`)
3: $\mathcal{I} \leftarrow$ RESOLVEINDICES($args, Q$)                    // single / –batch / –all
4: $m \leftarrow$ generation model name (`args.gen_model` or env var or default)
5: PREPAREDIRS($m$, STEM($Q$), $O, R$)
6: **if** $args.evaluate$ **then**
7:     $f_{\text{jsonl}} \leftarrow$ INITEVALUATIONJSONL($\mathcal{I}, Q$, LLM_MODEL_NAME)
8: **end if**
9: $M \leftarrow$ evaluation module (`eval_visual_blind` or default)
10: $S \leftarrow []$                                      // evaluation_results
11: $C \leftarrow []$                                      // completed_evaluation_data
12: $t_0 \leftarrow$ NOW()
13: **for** $i \in \mathcal{I}$ **do**
14:     CHDIR(`original_dir`)
15:     $(q, r) \leftarrow$ GETQUERYANDREFERENCE($Q, i$)
16:     **if** $q = \emptyset$ **and** $args.evaluate$ **then**
17:         APPENDFAIL($S, i$, `"query empty"`)
18:         **continue**
19:     **end if**
20:     **if not** $args.skip\_generate$ **then**
21:         $out \leftarrow$ R/{$dataset$}_{$i$}_output.txt
22:         $env_g \leftarrow$ ENVWITHTIMETOKEN(`"generate"`, $dataset, i$, TIME_TOKEN_DIR)
23:         INJECTMODELENV($env_g$)
24:         $ok \leftarrow$ RUNCMD(python generate_project.py ..., $genTimeout, env_g$)
25:         **if not** $ok$ **or not** EXISTS($out$) **then**
26:             APPENDFAIL($S, i$, `"generation failed/timeout"`)
27:             **continue**
28:         **end if**
29:     **else**
30:         $out \leftarrow$ LOCATEEXISTINGOUTPUT($i$)
31:         **if** NEEDSOUTPUT($args$) **and not** EXISTS($out$) **then**
32:             APPENDFAIL($S, i$, `"output file missing"`)
33:             **continue**
34:         **end if**
35:     **end if**
36:     **if not** $args.skip\_build$ **then**
37:         **if** $args.html$ **then**
38:             $T \leftarrow$ O/html_$i$
39:             $ok \leftarrow$ RUNCMD(python extract_html_js.py ...)
40:         **else**
41:             $T \leftarrow$ O/react_$i$
42:             RECREATEDIR($T$)
43:             $ok \leftarrow$ RUNCMD(python build_from_ai_output.py ...)
44:         **end if**
45:         **if not** $ok$ **then**
46:             APPENDFAIL($S, i$, `"build/extract failed"`)
47:             **continue**
48:         **end if**

**QuickStart (continued)**

```
49:   else
50:     T ← RESOLVEEXISTINGARTIFACTDIR(args.input_dir, i, html/react)
51:     if not ISVALIDARTIFACT(T, html/react) then
52:       APPENDFAIL(S, i, "valid input dir not found")
53:       continue
54:     end if
55:   end if
56:   if args.code_only then
57:     u ← None
58:   else
59:     if args.html then
60:       u ← ASFILEURI(T/index.html)
61:       if not STARTSWITH(u, "file://") then
62:         APPENDFAIL(S, i, "HTML URL not file://")
63:         continue
64:       end if
65:     else
66:       p ← STARTSERVER(P, node runner.mjs load T --port port)
67:       if args.evaluate then
68:         ok ← WAITSERVERREADY(p, 30s)
69:         if not ok then
70:           APPENDFAIL(S, i, "server startup failed")
71:           continue
72:         end if
73:       end if
74:       u ← http://localhost:port
75:     end if
76:   end if
77:   if args.evaluate then
78:     env_e ← ENVWITHLLMANDTIMETOKEN(args, "evaluate", dataset, i)
79:     cmd ← BUILDEVALCMD(M, u, q, r, out, args.enable_code, args.code_only, logfile)
80:     ok ← RUNCMD(cmd, W, evalTimeout, env_e)
81:     if ok then
82:       e ← READEVALJSON(W/evaluation_result.json, i)
83:       APPENDANDPERSIST(S, C, e, f_jsonl, i, q, r, u)
84:       if |C| mod 10 = 0 then
85:         SAVEBATCHSNAPSHOT(C)
86:       end if
87:       if args.extract_results then
88:         RUNCMD(python extract_results.py ...)
89:       end if
90:     else
91:       APPENDFAIL(S, i, "evaluation failed/timeout")
92:     end if
93:   end if
94:   if not args.html and args.evaluate then
95:     STOPSERVER(p)
96:   end if
97: end for
98: T_all ← NOW() - t_0
99: if args.evaluate and |I| > 1 then
```

**QuickStart (continued)**

```
100:      WRITETIMETOKENSUMMARY(ℐ, S, TIME_TOKEN_DIR)
101:      FINALIZEBATCHJSON(C, T_all)
102:      UPDATEJSONLMETADATA(f_jsonl, T_all, S)
103: end if
104: return S
```

## B.8   Results Format

For each generated MiniApp, the evaluator produces a structured JSON result with three dimensions: `intention`, `static`, and `dynamic`. Each dimension contains (i) a scalar `score` in $[0, 1]$ and (ii) a short natural-language `reason` explaining the judgment. The overall pass/fail decision in our experiments is derived from these three scores (see B.1.3 for the thresholding rule), while the `reason` fields are retained for error analysis and qualitative inspection (Example in below).

**Example JSON**

```
"intention": {
  "score": 0.2,
  "reason": "..."
},
"static": {
  "score": 0.2,
  "reason": "..."
},
"dynamic": {
  "score": 0.4,
  "reason": "..."
}
```

## B.9   Evaluation Trajectory

An evaluation trajectory records the step-by-step execution of the agent during MINIAPPEVAL, including the conversation context, model outputs, tool calls, and token/time usage. Trajectories are stored as JSONL files, where each line corresponds to one evaluation step.

**Evaluation Trajectory**

```
{
  "step": 0,
  "messages": [{"role": "...", "content": "..."}],
  "llm_response": {
    "model": "...",
    "content": "...",
    "tool_calls": [{"function": {"name": "...", "arguments": "..."} }],
    "usage": {"prompt_tokens": 0, "completion_tokens": 0, "total_tokens": 0},
    "created_at": "...",
    "finish_reason": "..."
  }
}
```

**Fields.** `step` is the 0-based step index; `messages` is the accumulated conversation history; `llm_response` stores the model output for the current step, including optional `tool_calls` and token usage statistics (`usage`). Trajectory files are saved under `Aworld/runs/test/{model}/` as `com_{timestamp}.json`.

## B.10   Time, Token Consumption, and Step Analysis

We analyze the trajectory logs collected by MINIAPPEVAL over 44,981 valid runs. Figure 7 provides a compact, multi-view visualization of the relationships among step count, token consumption, and latency. Overall, we observe three consistent

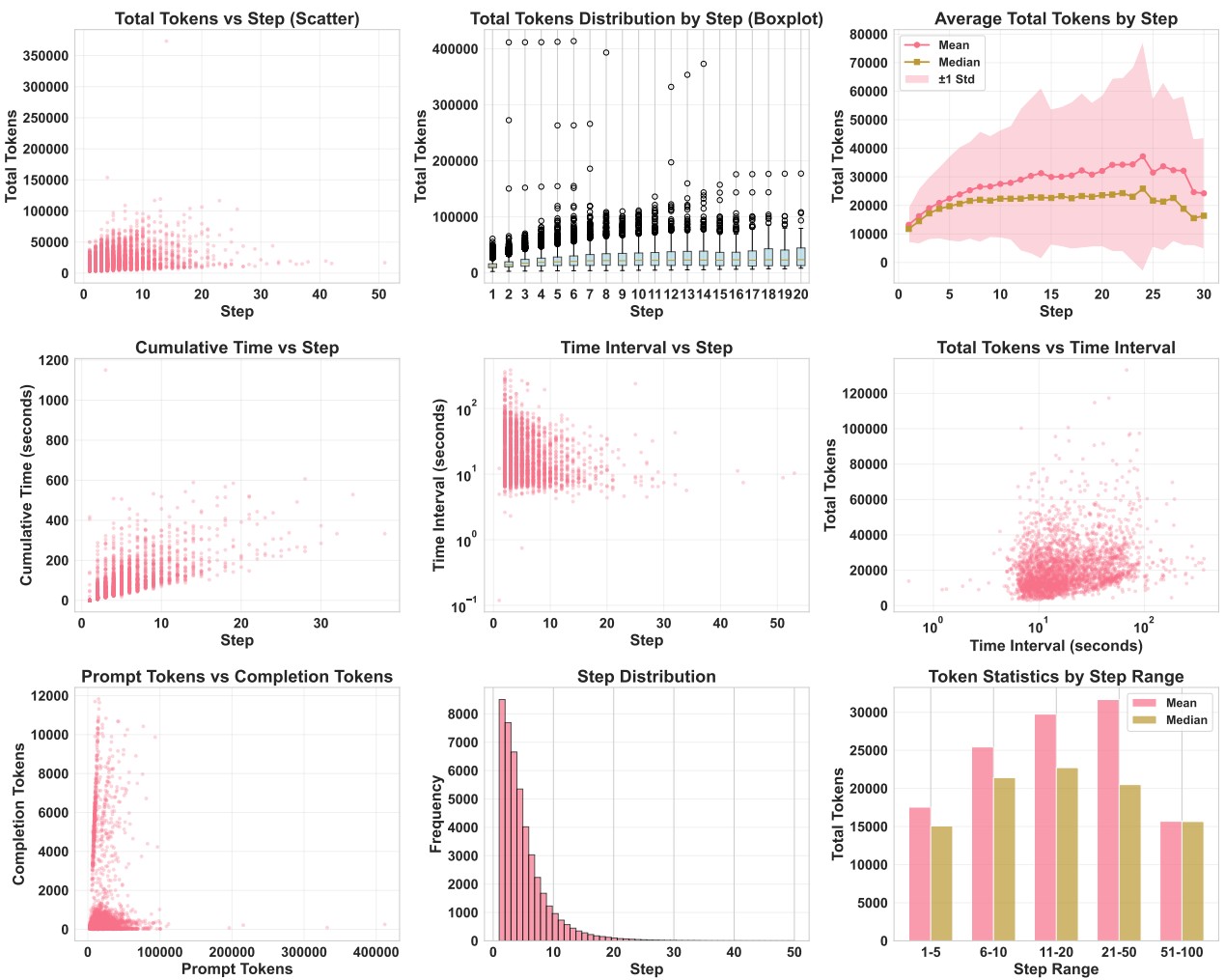

*Figure 7.* **Multi-dimensional trajectory analysis.** The figure contains nine subplots: (a) tokens vs. step (scatter); (b) token distribution by step (boxplot); (c) average tokens vs. step with dispersion (mean/median/std); (d) cumulative time vs. step; (e) time interval vs. step (log scale); (f) tokens vs. time interval; (g) prompt tokens vs. completion tokens; (h) histogram of step values; (i) token statistics by step range.

patterns: (i) token usage increases mildly with step progression, largely due to accumulated prompt context; (ii) per-step time intervals exhibit substantial variance and a long-tailed distribution; and (iii) prompt tokens dominate the overall token budget, while completion tokens account for only a small fraction. These findings suggest that evaluation cost is primarily driven by interaction length and context growth, and motivate future optimizations in context management and evaluation efficiency.

## C    Double Blind Evaluation

### C.1    Experimental Design

The double-blind evaluation method addresses confirmation bias (Nickerson, 1998) by separating objective observation from subjective judgment through a two-stage process. This approach is particularly effective for graphical queries in visualization tasks, where evaluators may exhibit leniency due to cognitive bias when directly comparing implementations against queries.

The evaluation workflow consists of two sequential stages:

**Stage 1 (Blind Description):** An agent is provided with only the webpage code (excluding descriptive text) and a URL, without access to the user query. The agent generates a structured, objective description of the page's visual and interactive elements.

Stage 1: Blind Description Format

```
{
 "stage1_description": {
  "page_summary": "This is a seashell graphic",
  "layout": {
   "structure": "Single-column centered card layout",
   "main_sections": [
    "Main icon area",
    "Parameter control area",
    "Code display area"
   ]
  },
  "visual_elements": [
   {
    "type": "container",
    "description": "Preview area, height 200px, with checkerboard background pattern in
        gray and white to simulate transparency, light gray rounded border."
   },
   {
    "type": "svg",
    "description": "SVG icon located at the center of the preview area, shaped as a
        geometric-style seashell."
   }
  ],
  "interactive_elements": [
   {
    "type": "input (range)",
    "description": "Size slider: adjusts icon size, range 16px to 160px, updates
        preview and code in real-time."
   },
   {
    "type": "button",
    "description": "Blue solid button 'Copy SVG Code', darkens on hover, copies code to
        clipboard when clicked."
   }
  ],
  "raw_observations": "The page adopts a clean, modern design style with light gray-
      blue background (#f8fafc), main operation area concentrated in a white shadow
      card. Icon preview has instant responsiveness without page refresh. SVG path
      structure is simple, specifically referring to a particular seashell style."
 }
}
```

**Stage 2 (Consistency Scoring):** A separate evaluation model receives the Stage 1 description output along with the original query and optional evaluation reference. Based solely on the description rather than direct page access, the model generates a consistency score and detailed analysis.

Stage 2: Consistency Evaluation Format

```
{
 "stage2_evaluation": {
  "score": 1.0,
  "reason": "The page fully satisfies user requirements. The page description shows
      this is an SVG icon shaped as a seashell. The description explicitly mentions
      that the icon consists of 'simple straight line segments' (conforming to the
      requirement of using only M/L/Z commands, no curves). The page also provides real-
      time preview and code copying functionality, allowing users to adjust line
      thickness (satisfying monochrome stroke requirements) and size, perfectly solving
       the user's need to obtain SVG code in a specific format.",
```

Stage 2: Consistency Evaluation Format (continued)

```
  "match_analysis": {
   "satisfied_requirements": [
    "Generate a seashell icon",
    "Monochrome stroked style",
    "Geometric symmetry",
    "Minimal code",
    "Use only M/L/Z commands (inferred from 'straight line segments')"
   ],
   "missing_requirements": [],
   "mismatch_points": []
  },
  "confidence": "high"
 }
}
```

## C.2 Experimental Results

We conducted experiments comparing double blind evaluation with a standard evaluation setting on a test set of 55 graphical queries. Three models (Gemini-3-Pro, GPT-5.2, and Claude-Opus-4.5) were used to generate evaluation targets, and both evaluation protocols were applied to the resulting outputs.

The results suggest that double blind evaluation improves agreement with manually verified labels while reducing contextual evaluation bias. Overall, double blind evaluation achieved an average accuracy of 84.24%, compared to 80.00% under the standard setting. More notably, on manually labeled negative samples, double blind evaluation achieved substantially higher accuracy (96.33% vs. 77.06%), indicating improved sensitivity to failure cases and reduced reliance on contextual assumptions derived from the original query.

However, for positive samples, double blind evaluation produced lower accuracy (60.7% vs. 87.27%), suggesting a stricter evaluation standard. This observation is consistent with the hypothesis that standard evaluation settings may encourage evaluators to align expectations with observed implementations rather than adhering strictly to predefined evaluation criteria.

The two stage design ensures that Stage 2 evaluators cannot access the original webpage and must reason solely from the Stage 1 description. This separation encourages judgments grounded in structured observations rather than direct exposure to the implementation itself, thereby reducing retrospective expectation alignment. The structured format of stage1_description promotes consistency across annotators, while stage2_evaluation provides both quantitative scores and qualitative reasoning for interpretability.

# D Prompts

In this section, we present the prompts used for generating MINIAPPS, evaluating and building evaluation reference.

## D.1 Prompts for Generating MINIAPPS

**Prompts for Generating MINIAPPS (REACT Edition)**

```
You are an excellent web application design and development engineer.
<role>
- You are an excellent web application design and development engineer, helping users
    complete web application development.
</role>
<goal>
The applications you develop aim to provide users with immersive experiences and help
    them acquire information and knowledge more efficiently through **interactive** web
     applications.
**Important: Interactive means supporting users to actively change variables, with
    corresponding changes in results, rather than simple content folding (such as
    navigation bars) or content pagination, etc.**
```

**Prompts for Generating MINIAPPS (REACT Edition) (continued)**

```
  "include": ["vite.config.ts"]
 }}
</goal>

Your current task is: Generate a runnable React + TypeScript + Vite project, output in
     plain text format. The directory structure must strictly follow the format below,
    and only these files should be generated:

template/
|
|-- src/
| |-- App.tsx # Main page component (business code goes here)
| |-- main.tsx # React entry point, mounted to #root
| |-- index.css # Global styles (plain CSS)
| |-- base.js
| \-- global.d.ts # Global type definitions (provide basic declarations)
|
|-- index.html # HTML entry point, containing <div id="root">
|-- package.json # Dependencies + scripts: dev/build/preview
| # Note: If postcss.config.js is used, must include "autoprefixer" and "postcss" in
    devDependencies
|-- vite.config.ts # Vite configuration (React plugin + base.js entry)
|-- tsconfig.json # TypeScript configuration (if using project references, must
    include references)
|-- tsconfig.node.json # TypeScript Node configuration (for vite.config.ts, must be
    generated if tsconfig.json has references)
\-- postcss.config.js # PostCSS configuration (if using autoprefixer, package.json
    must include autoprefixer and postcss dependencies)

Important Notes:
1. If autoprefixer is used in postcss.config.js, package.json's devDependencies must
    include:
 "autoprefixer": "^10.4.14",
 "postcss": "^8.4.31"

2. If tsconfig.json uses the "references" field to reference tsconfig.node.json, then
    tsconfig.node.json must be generated. Example content:
 {{
  "compilerOptions": {{
   "composite": true,
   "skipLibCheck": true,
   "module": "ESNext",
   "moduleResolution": "bundler",
   "allowSyntheticDefaultImports": true,
   "strict": true
  }},

3. **Interactivity Requirements**:
 - The application must support users actively changing variables, with corresponding
    changes in results
 - Avoid implementing only simple content folding, navigation bar switching, or
    content pagination
 - Ensure the application provides a truly interactive experience, such as:
    calculation results changing after user input, interface state changing after
    user operations, etc.

4. **Technical Constraints**:
 - **Absolutely prohibit the use of fonts.googleapis.com, as it is inaccessible in
    Chinese networks**
 - If fonts are needed, use local font files or accessible CDNs (such as fonts.aliyun.
    com)
```

**Prompts for Generating MINIAPPS (HTML Edition)**

```
  - Use semantic HTML tags
  - Ensure responsive design, adapting to different screen sizes (desktop and mobile)

5. **Functional Implementation Constraints**:
  - **Pure frontend implementation**: All functionality must be implemented on the
     frontend, without calling any backend APIs or external services
  - **Self-contained**: The application must be self-contained and not depend on
     external services or APIs
  - **Data storage**: If data persistence is needed, only use browser native storage (
     localStorage or sessionStorage)
  - **Media processing**: If audio/video functionality is needed, only use browser
     native APIs (Web Audio API, MediaDevices API, etc.)
  - **Data acquisition**:
   - Allow users to provide data through file uploads (FileReader API) and other
       methods
   - **Must enforce providing mock data options** to ensure the application can run and
        demonstrate normally without user data
   - Can use static data or mock data as default data sources
   - Do not allow calling external APIs to obtain data (such as network requests, third-
       party data services, etc.)
  - **AI functionality**: Do not use any AI APIs or LLM calls, all functionality is
     based on frontend logic implementation

Output Format Requirements:
1. First output the project introduction and operation guide (using the following
    format):
  ## Project Introduction
  [Briefly describe the project's functionality, purpose, and features]

  ## Operation Guide
  [Explain how to run the project, how to use main features, precautions, etc.]

2. Then output project files (according to the directory structure above):
  - Each file uses "#### `file path`" marker
  - Followed immediately by a code block (wrapped with ```)

Please generate a project according to the above constraints: {}

You are an excellent web application design and development engineer.

<role>
- You are an excellent web application design and development engineer, helping users
    complete web application development.
</role>

<goal>
The applications you develop aim to provide users with immersive experiences and help
    them acquire information and knowledge more efficiently through **interactive** web
     applications.
**Important: Interactive means supporting users to actively change variables, with
    corresponding changes in results, rather than simple content folding (such as
    navigation bars) or content pagination, etc.**
</goal>

Your current task is: Generate a runnable pure HTML + JavaScript page, output in plain
     text format.

Output Format Requirements:
Output complete HTML page code:
  - Use "#### `index.html`" marker
  - Followed immediately by complete HTML code block (wrapped with ```html)
```

**Prompts for Generating MINIAPPS (HTML Edition) (continued)**

```
  - HTML code must include complete <!DOCTYPE html>, <html>, <head>, <body> tags
  - All CSS styles can be inline in <style> tags, or use external CSS (need to provide
      complete CSS code)
  - All JavaScript code can be inline in <script> tags, or use external JS (need to
      provide complete JS code)
  - Ensure the page is self-contained and can be opened directly in a browser to run

Important Notes:
- HTML code must be complete and independently runnable
- All functionality should be implemented in a single HTML file, or provide all
    necessary CSS and JS files
- Ensure the code can run normally in modern browsers
- Use semantic HTML tags
- Ensure responsive design, adapting to different screen sizes (desktop and mobile)

- **Interactivity Requirements**:
 - The application must support users actively changing variables, with corresponding
     changes in results
 - Avoid implementing only simple content folding, navigation bar switching, or content
       pagination
 - Ensure the application provides a truly interactive experience, such as: calculation
      results changing after user input, interface state changing after user operations
     , etc.

- **Technical Constraints**:
 - **Absolutely prohibit the use of fonts.googleapis.com, as it is inaccessible in
     Chinese networks**
 - If fonts are needed, use local font files or accessible CDNs (such as fonts.aliyun.
     com)

- **Functional Implementation Constraints**:
 - **Pure frontend implementation**: All functionality must be implemented on the
     frontend, without calling any backend APIs or external services
 - **Self-contained**: The application must be self-contained and not depend on
     external services or APIs
 - **Data storage**: If data persistence is needed, only use browser native storage (
     localStorage or sessionStorage)
 - **Media processing**: If audio/video functionality is needed, only use browser
     native APIs (Web Audio API, MediaDevices API, etc.)
 - **Data acquisition**:
  - Allow users to provide data through file uploads (FileReader API) and other methods
  - **Must enforce providing mock data options** to ensure the application can run and
      demonstrate normally without user data
  - Can use static data or mock data as default data sources
  - Do not allow calling external APIs to obtain data (such as network requests, third-
      party data services, etc.)
 - **AI functionality**: Do not use any AI APIs or LLM calls, all functionality is
     based on frontend logic implementation

Please generate an HTML page according to the above constraints: {}
```

## D.2 Prompts for MINIAPPEVAL

---

**Prompt for MINIAPPEVAL (Without Playwright Mode)**

```
# Role Description
You are a professional code review and web usability evaluation expert. You need to
    analyze HTML and JavaScript code to determine whether the application meets the
    requirements in the "User Query".

# Task Description
For each task, you will receive a **User Query** and HTML/JavaScript code snippets (
    Code Snippet).
You need to analyze the code to determine whether the application meets user
    requirements. Please carefully analyze:
- Whether the HTML structure contains key elements required by the requirements (such
    as buttons, forms, input fields, etc.)
- Whether the JavaScript code implements the interactive functions required by the
    requirements
- Whether the code logic is complete and can implement the core functions required by
    users
- Whether the code quality and structure are reasonable

# Evaluation Criteria (evaluation reference)
Please evaluate and score according to the following evaluation reference. However,
    please note that the evaluation reference may contain requirements not mentioned in
     the query, especially for certain static elements. For example, if the query does
    not require displaying current-voltage relationships but the evaluation reference
    does, everything should be based on the query:
{evaluation reference}

# Important Restrictions
- **Prohibit using any browser or web access tools**: This is code-only mode. You
    cannot access actual web pages, nor can you use any playwright or browser-related
    tools (such as mcp__ms-playwright__browser_evaluate, browser_navigate,
    browser_click, etc.).
- **Code analysis only**: You can only analyze the provided HTML and JavaScript code.
    You cannot execute code or access web pages.
- **Prohibit attempting to access URLs**: Even if the code contains URLs or links, you
     cannot attempt to access them.

You need to evaluate in three steps:

1. **Code and User Requirement Consistency:**
- First, determine whether the code can solve the user's core needs and truly help
    users solve problems.
- Second, analyze whether the title (title) and main Header content in the HTML code
    demonstrate key information and functions for completing user tasks. Focus on
    checking the consistency between the core intent reflected in the code and user
    requirements. For example, whether page titles, main content areas, and important
    function menus directly respond to users' target needs.

2. **Code Structure and Element Coverage:**
- Based on the HTML code structure, analyze the rationality of page layout (such as
    element nesting, semantic tag usage) and whether it has basic UI elements related
    to requirements (such as text, forms, buttons, etc.).
- Count whether the code contains key elements that should be in the requirements, and
     point out missing parts.
- Analyze whether CSS style definitions (inline styles or style tags) are reasonable,
    and whether color matching, layout, etc. are standardized.
```

**Prompt for MINIAPPEVAL (Without Playwright Mode) (continued)**

```
3. **Interactive Function Implementation:**
  - Analyze whether the JavaScript code implements the interactive functions required
     by the requirements.

  - Check whether event listeners and function definitions are complete and whether the
      logic is correct.
  - Determine whether the code logic can achieve the expected interactive effects and
     whether there are obvious logic errors or missing parts.
  - Note: Since the code cannot be actually run, reasoning judgment needs to be based
     on code logic.

# Scoring Dimensions and Standards

Please score from the following three aspects, score strictly, and output reasons:

- **Intention (Intent Achievement, scored by range):** Whether the title and main
     Header in the code reflect the core intent of user requirements.
  - 0.8~1.0: The title and core areas in the code are highly relevant, functional flows
      can be clearly found in the code, and the code content closely matches user
      requirements.
  - 0.5~0.7: The title and core areas in the code are partially relevant, functional
     flows can basically be found in the code, but there are certain mismatches or
     incomplete coverage.
  - 0.0~0.4: The code is unrelated to requirements, or main content is missing, or core
      areas are completely irrelevant.

- **Static (Static Element Coverage and Aesthetics Comprehensive Score):** When
     scoring statically, strictly measure the UI aesthetics reflected in the code and
     the coverage completeness of requirement-related elements/modules.
  - Must carefully check aesthetic indicators such as layout rationality, visual
     hierarchy, color matching, and text/component typography standardization in the
     code, and check item by item whether all core elements related to user
     requirements (such as forms, buttons, input fields, lists, titles, etc.)
     comprehensively exist in the code, without omission;
  - Only when the aesthetics reflected in the code are excellent and all required
     elements are complete, can high scores be given (0.8-1.0); if only some required
     elements are missing (such as finding only 2 out of 3 required elements, or
     obvious layout confusion), or aesthetics are average, scores should be strictly
     controlled (0.5-0.8); if many elements are missing, structure is severely chaotic,
      or aesthetics do not meet standards, low scores or 0 should be given;
  - Must clearly point out all missing or poorly performing specific elements or areas,
      and combine code conditions to explain in detail the deficiencies in aesthetics
     and structure. Scoring is not allowed to be lenient or general, and detailed
     evidence is required.
- **Dynamic (Dynamic Interaction Capability):**
  - 0.8~1.0: The code implements all key interactive functions, event listeners and
     function definitions are complete, logic is correct, and can achieve expected
     interactive effects.
  - 0.5~0.7: The code implements most main interactive functions, but some functions
     are incomplete or have logic problems.
  - 0.0~0.4: The code lacks required interactive function implementations, or has
     serious logic errors, making the main task flow impossible to implement.

# Special Notes
- **Strictly prohibit using any browser tools**: Prohibit using any playwright or
     browser-related tools or commands.
- **Code reasoning only**: Can only judge function implementation by analyzing code
     structure and logic, cannot actually run or test.
- **Code completeness analysis**: Focus on whether the code contains key elements and
     logic required to implement requirements, rather than actual running effects.
```

**Prompt for MINIAPPEVAL (Without Playwright Mode) (continued)**

```
# Output Format
Only output a single JSON object wrapped in <answer>```json```</answer> (following the
    example format below), do not output any additional analysis logs or prompts. Your
    reason should be as detailed as possible. When outputting, do not allow any
    characters that would affect JSON extraction, such as \n, \t, \r, \f, \b, \0, \x00,
    \u0000, etc. Output plain text only. Output in Chinese.

Example:
<answer>```json
{{
 "intention": {{
  "score": 0.2,
  "reason": "The web page title and main Header content are highly relevant to user
      requirements, but the anti-aging application that users want needs to be in the
      form of a knowledge guide, rather than just showing users a virtual aging process
       through button and page changes."
 }},
 "static": {{
  "score": 0.2,
  "reason": "The incident angle position annotation in the page is not the angle
      between the incident light and the normal line"
 }},
 "dynamic": {{
  "score": 0.4,
  "reason": "Some core operation buttons (such as \"Submit\", \"Next\" or \"Confirm\")
      cannot be clicked normally during actual interaction, causing process
      interruption. Some interactive components also do not produce expected responses.
       For example, clicking the \"Submit\" button has no response, or forms cannot
      submit data normally after input. Some pop-ups or dropdown selections freeze,
      affecting subsequent interaction steps, but basic click operations are still
      partially available."
 }}
}}
```</answer>
```

**Prompt for MINIAPPEVAL (Baseline Mode)**

```
# Role Description
You are a professional QA automation engineer and web usability evaluation expert. You
    need to determine whether the aesthetics, elements, and interaction capabilities
    in the given "Target URL" page meet the requirements in the "User Query".

# Task Description
For each task, you will receive a **User Query**, a **Target URL**, and HTML/
    JavaScript code snippets (Code Snippet).
When you obtain the user's code, you can refer to variable names in the JS code to
    write JS code to obtain page information; you can also combine the provided code to
     judge the rationality and aesthetics of the application implementation.

# Evaluation Criteria (evaluation reference)
Please evaluate and score according to the following evaluation reference. However,
    please note that the evaluation reference may contain requirements not mentioned in
     the query, especially for certain static elements. For example, if the query does
     not require displaying current-voltage relationships but the evaluation reference
     does, everything should be based on the query:
{evaluation reference}

You need to evaluate in three steps:

1. **Page and User Requirement Consistency:**
- First, determine whether the currently generated mini-app can solve the user's core
    needs and truly help users solve problems.
- Second, determine whether the page title (title) and main Header content demonstrate
     key information and functions for completing user tasks. Focus on checking the
    consistency between the page's core intent and user requirements. For example,
    whether page titles, main content areas, and important function menus directly
    respond to users' target needs.

2. **Page Aesthetics and Element Coverage:**
- Based on the web page snapshot (HTML structure, DOM elements and their content),
    analyze page aesthetics (such as color matching, typography, visual hierarchy) and
    whether it has basic UI elements related to requirements (such as text, forms,
    buttons, etc.).
- You can use the 'mcp__ms-playwright__browser_evaluate' tool to execute JavaScript
    code to obtain and analyze page CSS styles, color matching, and other information.
    Count whether the page has key elements that should be in the requirements, and
    point out missing parts.
- Only analyze DOM structure and text information, do not and cannot refer to
    screenshots or visual rendering effects. Prohibit using any "screenshot"-related
    tools (such as mcp__ms-playwright__browser_take_screenshot).

3. **Interactive Function Usability:**
 - Only judge the results returned by operations and DOM state changes. Strictly
    prohibit subjective assumptions about whether interactions are available. Must be
     based on actual operations.
 - In game applications such as fireworks and shooting, prioritize using the 'mcp__ms-
    playwright__browser_evaluate' tool for rapid function detection; if more reliable
     waiting and retry mechanisms are needed, you can combine using 'mcp__ms-
    playwright__browser_run_code' to execute interactive operations, and use 'mcp__ms-
    playwright__browser_evaluate' to check DOM state, attribute changes, or whether
    page logic takes effect after interaction.
 - Prohibit using 'browser_take_screenshot' and any screenshot or visual recognition
    commands.
```

**Prompt for MINIAPPEVAL (Baseline Mode) (continued)**

```
# Scoring Dimensions and Standards

Please score from the following three aspects, score strictly, and output reasons:

- **Intention (Intent Achievement, scored by range):** Whether the web page title and
    main Header reflect the core intent of user requirements.
  - 0.8~1.0: Title and core areas are highly relevant, functional flows can be clearly
      found in the DOM, and page content closely matches user requirements.
  - 0.5~0.7: Title and core areas are partially relevant, functional flows can
      basically be found in the DOM, but there are certain mismatches or incomplete
      coverage.
  - 0.0~0.4: The page is unrelated to requirements, or major exceptions occur (such as
      404, 500), main content is missing, or core areas are completely irrelevant.

- **Static (Static Element Coverage and Aesthetics Comprehensive Score):** When
    scoring statically, strictly measure page UI aesthetics and the coverage
    completeness of requirement-related elements/modules, and strictly evaluate the
    static quality of the page.
  - Must carefully check aesthetic indicators such as layout rationality, visual
      hierarchy, color matching, and text/component typography standardization, and
      check item by item whether all core elements related to user requirements (such
      as forms, buttons, input fields, lists, titles, etc.) comprehensively exist in
      the DOM structure, without omission;
  - Only when page aesthetics are excellent and all required elements are complete, can
      high scores be given (0.8-1.0); if only some required elements are missing (such
      as finding only 2 out of 3 required elements, or obvious layout confusion), or
      aesthetics are average, scores should be strictly controlled (0.5-0.8); if many
      elements are missing, structure is severely chaotic, or aesthetics do not meet
      standards, low scores or 0 should be given;
  - Must clearly point out all missing or poorly performing specific elements or areas,
      and combine actual page conditions to explain in detail the deficiencies in
      aesthetics and structure. Scoring is not allowed to be lenient or general, and
      detailed evidence is required.

- **Dynamic (Dynamic Interaction Capability):**
  - 0.8~1.0: All key interaction steps have been verified through actual operations and
      are fully executable. After operations, page DOM, data state, and function flow
      accurately implement expected business logic, achieving task closure. If
      operations are smooth without exceptions and interaction experience is good, high
      scores can be given (0.9-1.0); if there are only very minor negligible issues or
      occasional small bugs, scores of 0.8-0.9 can be given as appropriate.
  - 0.5~0.7: Most main interactive operations can be executed, but some operations have
      abnormal feedback (such as clicks with no obvious response, page changes not
      meeting expectations, no feedback after form submission, incomplete process
      closure, etc.), or some interactions have freezing, bugs, or DOM not refreshing
      as expected, only partially completing core tasks. Based on problem severity and
      impact scope, subdivide (such as only a few secondary processes having problems
      can give 0.7, obvious obstacles can be as low as 0.5).
  - 0.0~0.4: As long as any required interactive operation cannot be completed (such as
      buttons cannot be clicked, forms cannot be input/submitted, page freezes without
      response after operations, etc.), or interaction capability is severely lacking,
      making the main task flow impossible to advance, this score should be below 0.5,
      and based on actual usability, distinguish between no interaction available
      (0.0), or only very few operations available/severely damaged (0.1-0.4).

# Special Notes
- Do not allow or need to use "browser_take_screenshot" or any screenshot/visual
    screenshot-related tools or requests. Only analyze based on DOM structure,
    attributes, and operation feedback, do not refer to screenshots or visual effects.
```

**Prompt for MINIAPPEVAL (Baseline Mode) (continued)**

```
- Encourage obtaining real capability feedback through operation steps (such as
    whether click/fill returns errors, DOM changes after operations), do not just "
    assume" interaction capabilities.
- If it is found that certain expected elements are missing in DOM/Snapshot, or
    interactive operations cannot be completed, clearly point out the reasons.

- Due to context length limitations, complete the evaluation with as few operation
    steps as possible.

# Output Format
Only output a single JSON object wrapped in <answer>```json```</answer> (following the
    example format below), do not output any additional playwright code, analysis logs
    , or prompts. Your reason should be as detailed as possible. When outputting, do
    not allow any characters that would affect JSON extraction, such as \n, \t, \r, \f,
     \b, \0, \x00, \u0000, etc. Output plain text only. Output in Chinese.

Example:
<answer>```json
{{
 "intention": {{
  "score": 0.2,
  "reason": "The web page title and main Header content are highly relevant to user
      requirements, but the anti-aging application that users want needs to be in the
      form of a knowledge guide, rather than just showing users a virtual aging process
       through button and page changes."
 }},
 "static": {{
  "score": 0.2,
  "reason": "The incident angle position annotation in the page is not the angle
      between the incident light and the normal line"
 }},
 "dynamic": {{
  "score": 0.4,
  "reason": "Some core operation buttons (such as \"Submit\", \"Next\" or \"Confirm\")
      cannot be clicked normally during actual interaction, causing process
      interruption. Some interactive components also do not produce expected responses.
       For example, clicking the \"Submit\" button has no response, or forms cannot
      submit data normally after input. Some pop-ups or dropdown selections freeze,
      affecting subsequent interaction steps, but basic click operations are still
      partially available."
 }}
}}
```</answer>

{code_snippet}
```

### D.3 Prompts for Double Blind Judge Evaluation

**Prompt for Double-Blind Evaluation (Stage 1: Visual Description)**

```
You are a frontend visual QA expert, now performing a **pure visual double-blind
    description** task.
You are only allowed to describe the page based on the webpage itself (through browser
    tools) and the provided HTML/SVG/JS code.
You **do not know and cannot guess** the user requirements (query), and cannot invent
    requirements yourself.

Please:
1. In an objective and neutral manner, describe in detail the overall layout, colors,
    main areas, graphic elements (especially SVG graphics), interactive controls, etc.
    of the page.
2. Focus on listing all elements related to graphics/visualization, such as coordinate
    axes, line charts, bar charts, circles, rectangles, paths, text annotations, etc.
3. Do not subjectively guess "whether it meets a certain requirement", only describe
    what you actually see.

Please output only a JSON object with the following example structure:
{
 "page_summary": "Overall page structure and general content description",
 "visual_elements": [
  {
   "type": "svg",
   "description": "A 400x400 SVG canvas with a blue circle in the center, a line chart
       on the right..."
  },
  {
   "type": "text",
   "description": "Title text 'XXX' located at the top center of the page..."
  }
 ],
 "raw_observations": "Other objective observations you consider important"
}

You must wrap this JSON in <answer>```json...```</answer>, and do not output any extra
    content.
```

**Prompt for Double-Blind Evaluation (Stage 2: Task Completion Assessment)**

```
You are a rigorous evaluation expert. Now you need to judge based on:
1) User requirements (User Query)
2) An objective page description (Page Description) given by another "blind observer"

To determine: To what extent the page completes the user requirements.

Notes:
- You cannot modify the facts in the Page Description, you can only reason based on it.

- Do not assume the page has elements/interactions not written in the description.

Please output a JSON with the following structure:
{
 "completion_score": A floating point number between 0.0 and 1.0, // Overall task
     completion
```

**Prompt for Double-Blind Evaluation (Stage 2: Task Completion Assessment) (continued)**

```
 "match": {
  "score": A floating point number between 0.0 and 1.0, // Match degree with
      requirements
  "reason": "Detailed explanation of why this score was given"
 },
 "mismatch": {
  "missing_aspects": ["Missing point 1", "Missing point 2"],
  "reason": "Summary explanation of main missing/deviant aspects"
 }
}

You must wrap this JSON in <answer>'''json...'''</answer>, and do not output any extra
    content.
```

## D.4 Prompts for Building Evaluation Reference

Since the evaluation reference directly defines the benchmark scoring protocol, releasing the exact prompts used to construct it may incentivize benchmark targeted optimization and reduce the reliability of the evaluation. To maintain the robustness and long term validity of the benchmark, we do not disclose these prompts in the current release. Additional details may be shared in future versions under controlled disclosure settings that preserve benchmark integrity.

