# OpenReview forum: "MiniAppBench: Evaluating the Shift from Text to Interactive HTML Responses in LLM-Powered Assistants"
_ICML.cc/2026/Conference — ICML 2026 spotlight_

### Official Review · Reviewer_y7Ei · 2026-02-25

**Soundness:** 3
**Presentation:** 3
**Significance:** 3
**Originality:** 3
**Overall Recommendation:** 4
**Confidence:** 4

**Summary:**

This paper tackles the challenge of evaluating LLMs on interactive web generation by introducing MINIAPPBENCH, a dataset of 500 real-world tasks. To effectively test these principle-driven "MiniApps," the authors built MINIAPPEVAL, an agent-based framework that uses Playwright to simulate human interaction and evaluate outputs across Intention, Static, and Dynamic dimensions. Through comprehensive testing—including ablation studies and a double-blind scoring protocol—the results highlight two key findings. First, today's LLMs still struggle significantly with these complex tasks; second, the proposed evaluation method closely mirrors expert human judgment.

**Compliance With Llm Reviewing Policy:**

Affirmed.

**Key Questions For Authors:**

1. Which specific LLM(s) serve as the evaluator backbone in MINIAPPEVAL, and how sensitive are the scores to the choice of judge model and decoding settings?
2. What is the stability of the evaluator—both intra-judge (same judge, different seeds) and inter-judge (different judge LLMs)? Can you report repeatability/CIs or a cross-judge agreement study?
3. The benchmark relies on a strict, uncalibrated threshold ($min(Intention, Static, Dynamic) > 0.8$). Where is the sensitivity analysis to prove the leaderboard rankings wouldn't drastically shift if this threshold were slightly relaxed (e.g., to 0.75) or if the scores were averaged instead of taking the minimum?

**Limitations:**

yes

**Strengths And Weaknesses:**

## Strengths

1. Novel Paradigm: Shifts the focus from static UI cloning and simple CRUD tasks to interactive, principle-driven "MiniApps".
2. Smart Evaluation Design: Solves the open-ended "ground truth" problem using a Playwright-driven agent that actively tests code across Intention, Static, and Dynamic dimensions.
3. Empirical Rigor: Features comprehensive benchmarking of major models, solid ablations, a clever double-blind judging setup for visual tasks, and high human alignment (kappa 0.81–0.89).


## Weaknesses

1. Opaque Evaluator Setup & Single-Judge Bias: The framework leans heavily on an LLM-as-a-judge, yet the specific identity and decoding settings of this evaluator are missing from the main text. The reliance on a single judge without cross-judge robustness checks leaves the benchmark vulnerable to backbone-specific biases and prompt hacking.

2. Unjustified Scoring & Difficulty Calibration: The exact mechanics of how interaction evidence translates into numeric scores—and why the >0.8 hard threshold was chosen—lack detailed operational justification.

3. Lack of Variance Analysis: The experimental results rely heavily on single-sample generations using default hyperparameters. The absence of deeper statistical checks—such as inter-judge variance, sampling variability, or "best-of-N" approaches—weakens the overall confidence and interpretability of the findings.

---

> ### Author Rebuttal · Authors · 2026-03-31
>
> We sincerely thank the reviewer for the detailed and encouraging review, and are glad the reviewer recognized the novel paradigm shift, smart evaluation      design, and empirical rigor of our work. We address each weakness below.
>
> ### Q1: for the decoding setting of judge model
>
> > The evaluator setup lacks transparency, and reliance on a single LLM judge raises concerns about bias and robustness.
>
> We apologize for not specifying the MINIAPPEVAL driver model in the main text. Gemini-3-Pro-Preview, noted in the anonymous submission, will be explicitly stated in the camera-ready version. It was selected after benchmarking a candidate set for best performance, and we also conducted bias analyses without finding any significant bias (see our response to Reviewer `bziQ`, Q1 for experimental details).
>
> For decoding, a temperature sensitivity study (threshold = 0.8) showed minimal effect on performance, as shown in Table 15. The paper uses the default **temperature = 1**, and the full decoding settings will be added in the camera-ready version.
>
> *Table 15. Performance of Evaluation Models Under Different Temperature Settings*
> |Temperature|GPT-5.2 (GT=43.5%)|||GLM-4.7 (GT=18.5%)|||Gemini-3-Pro-Preview (GT=26.5%)|||
> |-|-|-|-|-|-|-|-|-|-|
> ||JPass %|Acc %|F1 %|JPass %|Acc %|F1 %|JPass %|Acc %|F1 %|
> |0.5|54.5|89.0|88.8|20.5|97.0|92.3|35.5|91.0|85.5|
> |0.8|52.5|91.0|90.6|18.5|98.0|94.6|33.5|93.0|88.3|
> |1.0|46.0|92.3|92.4|18.5|94.5|94.5|28.0|90.7|90.7|
> |1.2|45.0|93.5|92.7|15.5|97.0|91.2|26.5|95.0|90.6|
>
> ### Q2: for the threshold weakness and the mechanics of scoring
>
> > The scoring mechanism and the choice of the 0.8 threshold lack clear justification.
>
> Regarding the exact mechanics of translating interaction evidence into numeric scores, we instructed the model to provide scores across three dimensions: Intention, Static, and Dynamic. The prompt (detailed in Appendix E.2) provides expert-designed, specific criteria for different score ranges, along with concrete examples, to guide the LLM in this translation process. We also analyzed ranking stability across thresholds, as detailed in our response to Reviewer `bziQ`, Q2, which further supports the robustness and scientific validity of our scoring mechanics.
>
> The threshold of 0.8 is established both theoretically and empirically. Theoretically, the expert guidance in our evaluation prompt divides the scoring spectrum into three bands: 0.8\~1.0, 0.5\~0.7, and 0.0\~0.4, making 0.8 a natural boundary. Empirically, we conducted a comprehensive evaluation by experimenting with different numerical thresholds (0.6, 0.7, 0.8, 0.9) and different aggregation methods for the multi-dimensional scores (namely, the average and the median), which confirmed that a threshold of 0.8 yields the best performance. For detailed experimental results, please refer to our response to Reviewer `vah4`, Q2.
>
> ### Q3: for lack of variance analysis
>
> > The results lack variance analysis, limiting confidence in their stability.
>
> To validate evaluation stability, we conducted experiments on a representative subset (see Reviewer `PXn5`, Q1, Stage 3 for subset representativeness).
>
> **Multiple generations from the same model:** We generated five outputs per model (GPT-5.2, GLM-4.7, Gemini-3-Pro-Preview) and evaluated each. Less than 10% of queries showed minor judgment fluctuations, and no query had fewer than 3/5 agreement.
>
> *Table 16. Pass Rate for 5 Generations from the Same Model*
> |Generation Model|Run 1 %|Run 2 %|Run 3 %|Run 4 %|Run 5 %|Std %|CV|
> |-|-|-|-|-|-|-|-|
> |GPT-5.2|46.0|48.5|46.0|49.0|49.0|1.58|0.0330|
> |GLM-4.7|18.5|17.5|18.5|16.5|17.5|0.84|0.0474|
> |Gemini-3-Pro-Preview|28.0|30.0|28.5|30.5|28.5|1.14|0.0393|
>
> *Table 17. Per-Query Consistency Analysis for 5 Generations from the Same Model*
> |Generation Model|5/5 Consensus %|≥4/5 Consensus %|Fleiss' κ|
> |-|-|-|-|
> |GPT-5.2|93.5|100|0.948|
> |GLM-4.7|94.5|99.5|0.924|
> |Gemini-3-Pro-Preview|89.5|100|0.905|
>
> **Multiple evaluations of the same sample:** Five repeated evaluations per sample showed high stability, with no sample having fewer than 4/5 agreement.
>
> *Table 18. Pass Rate for 5 Repeated Evaluations of the Same Sample*
> |Generation Model|Eval 1 %|Eval 2 %|Eval 3 %|Eval 4 %|Eval 5 %|Std %|CV|
> |-|-|-|-|-|-|-|-|
> |GPT-5.2|46.0|47.0|47.5|44.5|46.5|1.14|0.025|
> |GLM-4.7|18.5|18.5|18.0|18.0|17.5|0.42|0.023|
> |Gemini-3-Pro-Preview|28.0|29.0|27.5|28.5|30.0|1.00|0.035|
>
> *Table 19. Per-Query Consistency Analysis for 5 Repeated Evaluations of the Same Sample*
> |Generation Model|5/5 Consensus %|≥4/5 Consensus %|Fleiss' κ|
> |-|-|-|-|
> |GPT-5.2|95.0|100|0.960|
> |GLM-4.7|96.5|100|0.953|
> |Gemini-3-Pro-Preview|95.5|100|0.959|
>
> Both multiple generations and repeated evaluations confirm high consistency, supporting the stability and reliability of our evaluation framework.

---

> > ### Author Rebuttal · Reviewer_y7Ei · 2026-04-03
> >
> > Having received the requested information from the authors, which adequately supports the main text, I will continue to give a positive score.

---

> > > ### Author Response · Authors · 2026-04-03
> > >
> > > Thank you for your time and for maintaining your positive evaluation. We are glad that the additional information adequately addressed your concerns.

---

### Official Review · Reviewer_vah4 · 2026-03-12

**Soundness:** 3
**Presentation:** 3
**Significance:** 2
**Originality:** 3
**Overall Recommendation:** 4
**Confidence:** 3

**Summary:**

This paper introduces MINIAPPBENCH, a benchmark for evaluating LLMs on generating interactive HTML mini-applications. The benchmark contains 500 tasks across six domains, distilled from large-scale real-user queries through a multi-stage filtering and balancing pipeline. The paper also proposes MINIAPPEVAL, an agentic evaluation framework that combines code inspection with browser-based interaction to score outputs along Intention, Static, and Dynamic dimensions. Overall, the paper contributes a new benchmark setting, an automated evaluation pipeline, and an empirical study of current model capabilities on interactive application generation.

**Compliance With Llm Reviewing Policy:**

Affirmed.

**Key Questions For Authors:**

The paper can benefit from providing clear elaboration on the following aspects.
(1) The reliability claims for MINIAPPEVAL are currently supported by ablation and human-agreement studies on relatively limited subsets rather than the full 500-task benchmark. Could the authors provide a more fine-grained breakdown of evaluator reliability across domains and difficulty levels, or otherwise clarify why the 183-sample validation is sufficient to support the broader claim of stable evaluator behavior?
(2) Since all experiments are conducted in a sandboxed, self-contained HTML setting with fixed browser or runtime constraints, how should readers interpret the benchmark’s external validity for more realistic assistant scenarios involving backend APIs, multi-file applications, persistent state, or tool use? It would be helpful if the authors could clarify which claims are intended to generalize beyond the current controlled setting and which are not. A careful answer would help calibrate the practical significance of the benchmark.
(3) The main results depend on a specific decision rule: a MiniApp is counted as successful only when the minimum of the Intention, Static, and Dynamic scores exceeds 0.8. Could the authors report a sensitivity analysis showing whether model rankings and headline conclusions remain stable under alternative thresholds or aggregation rules, such as average-score criteria or different pass thresholds? This would clarify how much the reported pass rates depend on evaluator calibration rather than underlying model quality.

**Limitations:**

Yes.

**Strengths And Weaknesses:**

Soundness: (i) The ablation study is conducted on only 183 manually labeled samples, and the human agreement study also uses 183 items per model for three models, so the core reliability claim is ultimately validated on a relatively small subset compared with the full 500-query benchmark. This does not invalidate the method, but it does mean the paper has not yet shown that evaluator behavior is equally stable across all domains and difficulty levels. (ii) Some evaluation choices appear materially consequential but are not analyzed enough. For instance, the paper defines success using a threshold of 0.8 on the minimum of the Intention, Static, and Dynamic scores, yet there is not a sensitivity study showing whether model rankings remain stable if this threshold changes or if a different aggregation rule is used. Since a model can fail overall because of one weak dimension even when the other two are strong, this design choice could substantially affect the reported pass rates and deserves a more explicit robustness analysis.
•Presentation: (i) In this paper, some important implementation details are pushed into prompts and appendix material rather than being synthesized clearly in the main text. A concrete example is the Eval-Ref generation mechanism: the paper says the references are produced from general guidelines plus domain-specific instructions and then audited by experts, but the actual domain prompts span many highly specific instructions, which makes it hard for the reader to quickly judge how much evaluator behavior is driven by the benchmark design itself versus by handcrafted prompting. (ii) The novelty claims could be stated more precisely. For example, when the paper presents itself as introducing the first benchmark for this setting and a reliable evaluation framework, it would help to separate more carefully what is new at the dataset level from what is an engineering integration of existing judge, browser, and prompt-based components. (iii) While anonymous repositories are allowed by ICML, the repository in Github should not describe the submission as an ICML paper before acceptance. The citation entry should therefore be revised to remove the explicit ICML venue claim and use a neutral anonymized format during review.
•Significance: (i) The practical scope of the benchmark is narrower than some of the broader framing suggests. A concrete example is that all evaluations are conducted in a sandboxed, self-contained HTML setting with headless Chromium and fixed rendering constraints, which is sensible for control but also means the conclusions may not transfer directly to real assistant scenarios involving backend APIs, multi-file projects, persistent state, or external tool use. (ii) The paper could also do more to demonstrate that benchmark conclusions are robust enough to guide future model development. Since the final dataset contains 500 tasks selected after filtering and stratification from a much larger pool, the community value would be even clearer if the paper provided stronger evidence that the chosen subset does not materially privilege the current evaluation pipeline or particular model families.

---

> ### Author Rebuttal · Authors · 2026-03-31
>
> We sincerely thank the reviewer for the careful and constructive review. We provide detailed responses to each concern below.
>
> ### Q1: for the selection of the experimental dataset weakness
>
> > The subset (183 samples) limitation.
>
> Due to high annotation cost, we conducted ablation and human-agreement studies on a sampled subset. Its representativeness of the full dataset has been experimentally validated, supporting the generalizability of our conclusions. For further details, refer to our response to Reviewer `PXn5`, Q3.
>
> ### Q2: for threshold weakness
>
> > The decision rule (min score > 0.8) lacks sensitivity analysis, and its impact on results is unclear.
>
> We thank the reviewer for this question. Theoretically, our evaluation prompt (App. E.2) segments the scoring range into three bands—0.8–1.0, 0.5–0.7, and     0.0–0.4—making 0.8 a natural cutoff. Empirically, inter-annotator agreement peaks at T = 0.8 across tested thresholds (see Table 9 Reviewer `bziQ`, Q1). Alternative criteria using mean or median scores (Tables 12–13) yield lower optimal F1 than the stricter minimum-score method, reflecting the "weakest link" effect: a low score in any single dimension disqualifies the output.
>
> *Table 12. Human Agreement Using Average Score Criterion.*
> |Thresholds|GLM-4.7||GPT-5.2||Gemini-3-Pro-Preview||
> |-|-|-|-|-|-|-|
> ||Acc %|F1 %|Acc %|F1 %|Acc %|F1 %|
> |0.6|52.6|54.1|69.3|80.0|62.9|74.0|
> |0.7|61.7|56.3|72.5|81.6|72.6|79.1|
> |0.8|73.4|63.7|81.7|86.8|79.2|81.3|
> |0.9|81.8|65.9|84.3|87.1|75.1|70.3|
>
> *Table 13. Human Agreement Using Median Score Criterion.*
> |Thresholds|GLM-4.7||GPT-5.2||Gemini-3-Pro-Preview||
> |-|-|-|-|-|-|-|
> ||Acc %|F1 %|Acc %|F1 %|Acc %|F1 %|
> |0.6|57.1|54.8|72.5|81.6|69.5|77.6|
> |0.7|59.1|54.0|76.5|83.8|76.1|80.5|
> |0.8|69.5|61.2|81.0|86.4|76.1|78.7|
> |0.9|76.0|61.9|79.7|83.9|69.5|66.7|
>
>
> ### Q3: for insufficient synthesis of key implementation details weakness
>
> > Key implementation details are buried in prompts/appendix, making it unclear how much behavior is driven by design vs. prompting.
>
> We will summarize the Eval-Ref prompt structure and domain-specific instructions in the main text, clarifying their role in shaping evaluator behavior. We will address similar issues in the same manner.
>
>
> ### Q4: for novelty presentation weakness
>
> >The novelty claims are unclear, particularly in separating dataset contributions from evaluation framework integration.
>
> At the **dataset level**, we introduce HTML as a new LLM–user interaction form, with key properties of **customized interaction** and **real-world principles**. At the **evaluation level**, we propose an **evaluation reference** mechanism extending checkpoint-style evaluation by granting the judge greater autonomy, better suiting highly customized scenarios.
>
> ### Q5: for anonymous link
>
> >The anonymized GitHub repository improperly references ICML, which could compromise submission anonymity.
>
> We thank the reviewer for raising this point. Our repository is fully anonymous and private, with no ICML citation. The only mention of ICML was a statement describing this as "an anonymous submission to ICML 2026," which is consistent with the double-blind policy. We have nonetheless removed it to fully address the concern.
>
> ### Q6: for the scope of benchmark weakness
>
> > The benchmark is evaluated in a sandbox, and its external validity to more realistic assistant settings is unclear.
>
> We acknowledge that scenarios involving backend APIs, persistent state, and external tool use are beyond our current scope, and will clarify this in the Limitations section.
>
> A sandboxed setting is well-suited to our focus on customized interaction and real-world principles, as it avoids external dependencies and enables stable, fair model comparison.
>
> To assess robustness beyond single-file HTML, we additionally evaluate a React (multi-file) setting (Sec. 4.1, App. C.1.2). As shown in Table 14, while absolute pass rates decrease due to increased implementation complexity, relative model rankings remain consistent.
>
> *Table 14. HTML vs. React Pass Rates (200-query subset; see Reviewer PXn5, Q1).*
> |Mode|GPT-5.2 %|GLM-4.7 %|Gemini-3-Pro-Preview %|
> |---|---|---|---|
> |HTML|46.0|18.5|28.0|
> |React|37.5|16.5|20.0|
>
>
> ### Q7: for the datasets weakness
>
> >It is unclear whether the 500-task subset is representative or introduces bias toward certain models or the evaluation pipeline.
>
> The 500-task dataset results from a rigorous multi-stage filtering process (3,234 → 1,521 → 1,123 candidates), followed by stratified sampling across domain and difficulty dimensions, ensuring balanced and principled coverage (see Reviewer `PXn5`, Q1–Q2 for details and distribution tables).
>
>
> To address potential model bias, dataset construction involves multiple model families (Gemini-3-Pro-Preview, Claude Sonnet 4.5, GPT-5.2) jointly performing key stages: filtering, sampling, and difficulty stratification (Sec. 3.4). We find no evidence of evaluator bias toward same-family models also (see Reviewer `bziQ`, Q2).

---

> > ### Author Rebuttal · Reviewer_vah4 · 2026-04-02
> >
> > the rebuttle addressed my concerns.

---

> > > ### Author Response · Authors · 2026-04-02
> > >
> > > We are glad to hear that the rebuttal addressed your concerns. If the additional information has sufficiently resolved the issues raised, we would greatly appreciate your consideration of whether an updated score might be warranted.

---

### Official Review · Reviewer_bziQ · 2026-03-12

**Soundness:** 3
**Presentation:** 3
**Significance:** 3
**Originality:** 3
**Overall Recommendation:** 5
**Confidence:** 4

**Summary:**

This authors propose MiniAppBench, a benchmark of 500 tasks for evaluating LLMs' ability to generate interactive HTML applications from text queries.
The benchmark is constructed with a 4-stage pipeline: 1) filtering millions of real user queries down to 3234 candidates, 1) augmenting them with LLM, 3) anchoring each with structured evaluation references across 3 dimensions (Intention, Static, Dynamic), and 4) using stratified sampling to 500 tasks across 6 domains and 3 difficulty levels. The paper also proposes MiniAppEval, an evaluation framework that uses Playwright to interact with the generated apps and score them.
Experiments across 16 models show the GPT-5.2 achieves only ~45% pass rate, and MiniAppEval achieves Cohen's kappa of 0.81 - 0.89 against human judgments.

**Compliance With Llm Reviewing Policy:**

Affirmed.

**Key Questions For Authors:**

- What's the judge model used in the main paper and would this change if a different model is used (i.e., judge sensitivity)
- How sensitive are model rankings to the pass-rate threshold?

**Limitations:**

yes

**Strengths And Weaknesses:**

## Strength
- The paper is tackling an increasingly important question as the model's ability continue to improve.
- The combination of static code inspection, Playwright-based dynamic interaction, and structured evaluation references is well-motivated.
- The ablation study (Table 2) demonstrates that each component contributes: removing the agent / code access / eval-refs all reduce precision dramatically
- The evaluation is comprehensive and thoughtful. I appreciate the amount of models tested, human agreement testsm and the ablations on LLM judge (double blind).

## Weakness
- LLM as Judge is a fair evaluation setup but more analysis on this would remove risk of bias. The authors don't seem to specify the exact judge used. The model might preferentially score models of the same family.
- Minor: the figures are a bit dense and hard to parse

---

> ### Author Rebuttal · Authors · 2026-03-31
>
> We sincerely thank the reviewer for the thorough and positive feedback, and for the strong overall recommendation. We address the remaining concerns below.
>
> ### Q1: for the setting of evaluation model weakness
>
> >The setting of evaluation model is not clear.
> >If the model scores higher when evaluating models from the same family.
>
> Sorry for the inconvenience. The anonymous link specifies that the model used is Gemini3-Pro-Preview, and we will include this in the main text for the camera-ready version. We selected it after benchmarking five models—**Gemini3-Pro-Preview, GPT-5.2, Claude-Sonnet-4.5, Qwen3-Coder-480B-A35B-Instruct, and GLM-4.7**—and found it achieves the most stable performance and highest agreement with human annotations (average F1 92.4%) on a 200-sample subset (representativeness of the full dataset is discussed in our response to Reviewer `PXn5`, Q1 Stage 3).
>
> Regarding potential self-bias, Gemini3-Pro-Preview scored itself at 28.0% (1.5 points above ground truth), GPT‑5.2 at 46.0% (2.5 points above), and GLM‑4.7 at 18.5% (matching ground truth). These results show no evidence of undue favoritism.
>
> *Table 5. Judge: GPT-5.2 (best average F1 60.3%, threshold 0.6)*
> |T|GPT-5.2| |GLM-4.7| |Gemini-3-Pro-Preview| |
> |-|-|-|-|-|-|-|
> | |JPass %|Acc %|F1 %|JPass %|Acc %|F1 %|JPass %|Acc %|F1 %|
> |0.6|35.5|73.0|65.8|28.0|78.5|53.8|30.5|78.0|61.4|
> |0.7|25.0|73.5|61.3|16.5|91.0|74.3|8.5|76.0|31.4|
> |0.8|11.0|67.5|40.4|4.5|83.0|26.1|7.0|75.5|26.9|
> |0.9|0.0|56.5|-|0.0|81.5|-|0.0|73.5|-|
>
> *Table 6. Judge: Claude-Sonnet-4.5 (best average F1 65.3%, threshold 0.7)*
> |T|GPT-5.2| |GLM-4.7| |Gemini-3-Pro-Preview| |
> |-|-|-|-|-|-|-|
> | |JPass %|Acc %|F1 %|JPass %|Acc %|F1 %|JPass %|Acc %|F1 %|
> |0.6|53.5|70.0|69.1|43.5|78.0|68.6|48.5|70.0|55.2|
> |0.7|42.5|82.0|79.1|20.0|76.5|49.5|36.5|82.0|67.3|
> |0.8|28.5|85.0|79.2|9.0|76.5|33.8|21.5|91.0|77.5|
> |0.9|23.5|80.0|70.1|2.0|74.5|10.5|6.5|87.0|48.0|
>
> *Table 7. Judge: GLM-4.7 (best average F1 66.7%, threshold 0.7)*
> |T|GPT-5.2| |GLM-4.7| |Gemini-3-Pro-Preview| |
> |-|-|-|-|-|-|-|
> | |JPass %|Acc %|F1 %|JPass %|Acc %|F1 %|JPass %|Acc %|F1 %|
> |0.6|77.0|66.5|72.2|48.0|66.5|49.6|67.0|59.5|56.7|
> |0.7|66.0|77.5|79.5|41.0|72.5|53.8|53.0|73.5|66.7|
> |0.8|58.0|85.5|85.7|29.0|72.5|42.1|32.0|75.5|58.1|
> |0.9|28.0|74.5|64.3|10.0|79.5|28.1|19.5|77.0|50.0|
>
> *Table 8. Judge: Qwen3-Coder-480B-A35B-Instruct (best average F1 68.7%, threshold 0.8)*
> |T|GPT-5.2| |GLM-4.7| |Gemini-3-Pro-Preview| |
> |-|-|-|-|-|-|-|
> | |JPass %|Acc %|F1 %|JPass %|Acc %|F1 %|JPass %|Acc %|F1 %|
> |0.6|85.5|58.0|67.4|55.5|63.0|50.0|62.0|64.5|59.9|
> |0.7|81.5|62.0|69.6|50.5|68.0|53.6|55.0|71.5|65.0|
> |0.8|75.5|68.0|73.1|39.0|79.5|64.3|50.5|76.0|68.8|
> |0.9|59.5|84.0|84.5|21.5|83.0|57.5|31.5|67.0|43.1|
>
> *Table 9. Judge: Gemini-3-Pro-Preview (best average F1 92.4%, threshold 0.8)*
> |T|GPT-5.2| |GLM-4.7| |Gemini-3-Pro-Preview| |
> |-|-|-|-|-|-|-|
> | |JPass %|Acc %|F1 %|JPass %|Acc %|F1 %|JPass %|Acc %|F1 %|
> |0.6|57.5|85.6|89.1|38.5|77.3|65.3|45.5|89.8|90.0|
> |0.7|55.0|88.2|90.6|36.0|81.8|68.2|41.5|80.2|78.0|
> |0.8|46.0|92.3|92.4|18.5|94.5|94.5|28.0|90.7|90.7|
> |0.9|32.0|68.6|68.8|11.5|79.9|52.3|23.0|58.4|36.9|
>
> ### Q2: for threshold weakness
> > How sensitive are model rankings to the threshold?
>
> The model rankings remain highly stable across thresholds of 0.6, 0.7, 0.8, and 0.9, indicating robustness.
>
> Specifically, the Spearman's ρ between the rankings at any of the three other thresholds and the ranking at T=0.8 is greater than 0.97, and the Kendall's τ is no less than 0.90. The maximum shift in rank position for any model was only 2 spots. The detailed performance is as follows:
>
> *Table 10. Rankings Across Different Thresholds.*
> |Model|Pass Rate % (Rank)||||
> |-|-|-|-|-|
> |Threshold|0.6|0.7|0.8|0.9|
> |GPT-5.2|71.1(1)|62.0(1)|45.5(1)|35.4(1)|
> |Claude-Opus-4.5|62.0(2)|52.6(2)|41.1(2)|23.1(2)|
> |GPT-5.1|55.2(3)|45.5(3)|32.0(3)|21.1(3)|
> |Gemini-3-Pro-Preview|47.1(4)|36.8(4)|27.5(4)|13.8(5)|
> |Claude-Sonnet-4.5|42.4(5)|33.6(5)|26.4(5)|14.2(4)|
> |GLM-4.7|35.5(7)|26.8(8)|18.3(6)|9.3(7)|
> |Gemini-3-Flash|41.5(6)|29.8(6)|17.6(7)|8.5(8)|
> |MiniMax-M2.1|34.3(8)|27.2(7)|17.2(8)|10.2(6)|
> |Grok-4.1-Fast-Reasoning|29.4(9)|22.4(9)|13.7(9)|6.6(9)|
> |Mimo-V2-Flash|27.9(10)|18.9(10)|12.5(10)|6.2(10)|
> |GLM-4.5-Air|16.9(11)|11.3(11)|7.1(11)|2.4(12)|
> |Kimi-K2-Instruct|13.3(12)|8.9(12)|6.2(12)|2.5(11)|
> |Qwen3-235B-A22B|8.4(13)|4.4(13)|2.9(13)|1.1(15)|
> |Hunyuan-Turbos-Latest|9.4(14)|5.3(15)|2.3(14)|1.7(13)|
> |Qwen3-Coder-480B-A35B-Instruct|7.3(15)|4.8(14)|1.8(15)|1.4(14)|
> |Qwen3-32B|3.5(16)|2.4(16)|0.7(16)|0.2(16)|
>
>
> *Table 11. Analysis of Ranking Stability: Correlation with Rankings at T=0.8.*
> | |Spearman ρ|Kendall τ|Maximum Rank Difference|
> |-|-|-|-|
> |T=0.6 vs T=0.8|0.9941|0.9667|1|
> |T=0.7 vs T=0.8|0.9824|0.9333|2|
> |T=0.9 vs T=0.8|0.9765|0.9000|2|
>
>
> ### Q3: for figures weakness
>
> >Minor: the figures are a bit dense and hard to parse
>
> We will clarify the figures in the camera-ready version by splitting complex charts into smaller, detailed subfigures in the appendix.

---

> > ### Author Rebuttal · Reviewer_bziQ · 2026-04-02
> >
> > The authors have provided the information I requested, which adequately supplemented the main text. I will keep my positive score.

---

> > > ### Author Response · Authors · 2026-04-02
> > >
> > > Thank you for your time in reviewing our rebuttal and for the positive feedback. We are glad that the additional information adequately addressed your concerns.

---

### Official Review · Reviewer_PXn5 · 2026-03-13

**Soundness:** 3
**Presentation:** 3
**Significance:** 3
**Originality:** 3
**Overall Recommendation:** 4
**Confidence:** 3

**Summary:**

This paper introduces MiniAppBench, a benchmark of 500 tasks across six domains (Games, Science, Tools, Humanities, Visualization, Lifestyle) for evaluating LLMs' ability to generate interactive, principle-driven HTML applications (termed MiniApps). Unlike prior code generation benchmarks that focus on algorithmic correctness or static layout fidelity, MiniAppBench targets a richer setting where generated artifacts must embed real-world principles (e.g., physical laws, commonsense constraints) and support customized user interaction.

The authors also propose MiniAppEval, an agentic evaluation framework that combines static code inspection with dynamic browser-based testing via Playwright to assess generated applications along three dimensions: Intention, Static, and Dynamic. Evaluation of 16 models (open-source and closed-source) reveals that even the best model (GPT-5.2) achieves only a 45.46% pass rate, with an overall mean of 17.05%. The framework's reliability is validated through ablation studies, a double-blind evaluation protocol, and a human agreement study yielding Cohen's kappa of 0.81-0.89.

**Compliance With Llm Reviewing Policy:**

Affirmed.

**Final Justification:**

The rebuttal adequately complements the paper and addresses my concerns.

**Key Questions For Authors:**

1. Threshold Sensitivity: How sensitive are the pass rates and model rankings in Table 1 to the 0.8 threshold? Reporting results at 0.6, 0.7, and 0.9 would demonstrate robustness. Would the relative ordering of models change at different thresholds?

2. Dataset Composition Transparency: What fraction of the final 500 benchmark tasks are direct real-world user queries (from Stage 1) vs. LLM-augmented variants (from Stage 2)? This distinction matters for the "10M+ real-world queries" provenance claim.

**Limitations:**

Several limitations identified through analysis are not acknowledged:

1. The 0.8 pass/fail threshold is used without sensitivity analysis or justification.

2. The ablation uses only 183 samples.

**Strengths And Weaknesses:**

Strengths:

Soundness:

a. Comprehensive Validation:

The evaluation framework is rigorously validated from multiple angles: the ablation study demonstrates that all three components (Eval-Ref, Code, Playwright agent) contribute meaningfully, with removing the Playwright agent causing a 70.97pp precision drop. A human agreement study shows Cohen's kappa of 0.81-0.89 across three model tiers, and a double-blind protocol addresses confirmation bias in visual tasks.

b. Discriminative Power:

The benchmark provides meaningful performance stratification. Pass rates span from 0.66% (Qwen3-32B) to 45.46% (GPT-5.2), with clear gradients across difficulty levels (Easy: 34.05%, Mid: 13.89%, Hard: 4.34%) and a consistent open-source vs. closed-source gap.

Presentation:

a. The paper is well-structured with a clear progression from problem motivation through benchmark design to evaluation methodology and experiments. The appendix is thorough, providing detailed prompts, environment setup, and pipeline pseudocode.

Siginificance:

a. Timely and Relevant:

The benchmark addresses a genuine and timely gap. As LLM-powered applications increasingly generate interactive web artifacts (e.g., Claude Artifacts, ChatGPT Canvas), a principled evaluation methodology for such outputs is needed. The three-dimensional evaluation (Intention, Static, Dynamic) offers a more holistic assessment than prior screenshot-comparison or script-based approaches.

Weaknesses:

Soundness:

a. Inter-annotator agreement for Dataset Curation:

No inter-annotator agreement is reported for the manual curation steps in the dataset construction pipeline (Stages 1 and 3). While IAA is reported for the evaluation study, the quality of the benchmark tasks themselves lacks quantitative quality assurance.

b. Dataset Provenance:

The dataset provenance claim of "10M+ real-world user queries" is partially diluted by Stage 2's LLM-assisted augmentation, which expands 1,123 seed queries to 1,974 candidates. The proportion of final 500 tasks originating from genuine user queries vs. synthetic variants is not disclosed.

Sigiificance:

The ablation study and human agreement study each use only 183 samples, which is modest relative to the full 500-task benchmark. Expanding validation to a larger subset would strengthen confidence in the framework's reliability.

---

> ### Author Rebuttal · Authors · 2026-03-31
>
> We sincerely thank the reviewer for the thorough feedback and are glad the reviewer recognized our benchmark's soundness, discriminative power, and relevance. We address each concern below.
>
> ### Q1: for inter-annotator agreement weekness:
>
> > Lack of IAA reporting for manual curation stages (Stages 1 & 3).
>
> We supplement the missing IAA: **Fleiss' κ = 0.75** (LLM filtering) and **κ = 0.87** (expert review), both indicating substantial agreement. These will be included in the camera-ready.
>
> **Stage 1** (Sec. 3.4): Rule-based cleaning yielded **3,234 candidates**. Three LLMs (GPT-5.2, Gemini-3-Pro-Preview, Claude Sonnet 4.5) then independently assigned binary removal labels; only unanimously flagged items were discarded, leaving **1,521 queries** (κ = 0.75). Three human experts re-screened these under the same protocol, retaining **1,123 items** (κ = 0.87). Domain assignment followed the ERA framework, as LLMs tend to under-account for real-world principles in categorization (App. B.2.2).
>
> **Stage 3**: The same three models generated code for candidate queries. Difficulty labels were assigned from execution outcomes, then combined with domain labels for stratified sampling to form the final **500-item dataset**. A **200-item subset** with matched distributions was constructed for expert annotation and ground truth production. All ablations and human agreement studies were conducted on this subset (or its **183-item derivative**).
>
> *Table 1. Domain Distribution (200 vs. 500 items).*
> |Domain|200 items(%)|500 items(%)|
> |-|-|-|
> |Science|35.0|37.4|
> |Games|23.0|24.2|
> |Tools|13.0|11.4|
> |Humanities|8.0|9.4|
> |Lifestyle|7.5|6.4|
> |Visualization|13.5|11.2|
>
> *Table 2. Difficulty Distribution (200 vs. 500 items).*
> |Level|200 items(%)|500 items(%)|
> |-|-|-|
> |easy|28.5|29.4|
> |medium|42.0|39.2|
> |hard|29.5|31.4|
>
> *Table 3. Part of Models' Pass Rates Comparison (200 vs. 500 items).*
> |Model|PassRate(500) (%)|PassRate(200) (%)|Δ (%)|
> |-----|------------|------------|--|
> |GPT-5.2|45.46|46.00|+0.54|
> |Claude-Opus-4-5|41.14|41.50|+0.36|
> |Gemini-3-Pro-Preview|27.52|28.00|+0.48|
> |GLM-4.7|18.31|18.50|+0.19|
> |Qwen3-32B|0.66|0.50|-0.16|
>
> ### Q2: for dataset weekness:
>
> >The proportion of final benchmark tasks derived from real user queries vs. LLM-augmented variants is unclear.
>
> We clarify the composition of the final benchmark. **15.4% of tasks (77/500)** are drawn directly from real online user queries, while **84.6% (423/500)** are LLM-augmented variants, all based on genuine user seeds that preserve the original topic and intent.
>
> Raw online queries are heavily skewed toward simple requests and popular domains, and often contain user-sensitive information. We therefore employ an LLM-driven evolutionary augmentation process to derive variants that preserve each seed's topic and intent while expanding difficulty and domain coverage. As shown in Table 4, this yields balanced coverage and improves discriminative power across models. The higher proportion of augmented queries thus reflects a design choice for balance, not weak real-world grounding.
>
> *Table 4: Domain × Difficulty Level Distribution (500 tasks)*
> |Domain|Easy|Medium|Hard|Total|
> |-|-|-|-|-|
> |Science|46(25%)|72(38%)|69(37%)|187|
> |Games|23(19%)|49(40%)|49(40%)|121|
> |Tools|13(23%)|27(47%)|17(30%)|57|
> |Visualization|30(54%)|20(36%)|6(11%)|56|
> |Humanities|16(34%)|18(38%)|13(28%)|47|
> |Lifestyle|19(59%)|10(31%)|3(9%)|32|
> |**Total**|**147(29%)**|**196(39%)**|**157(31%)**|**500**|
>
> The example below illustrates the seed-to-augmentation relationship:
>
> **Seed**: "Simulate the solar system."
>
> **Augmented**: "Create a solar system planetary simulator showing a planet on a flattened elliptical orbit. Allow users to shade the sector area swept over a time interval, verifying that equal time intervals sweep equal areas. Display the planet's instantaneous velocity vector in real-time."
>
>
> ### Q3: for the selection of the experimental dataset weakness
>
> > Validation is conducted on only 183 samples, which is small relative to the 500-task benchmark and may limit confidence in reliability.
>
> Due to high annotation cost, ablation and human agreement studies were conducted on the 200-item subset (Q1 Stage 3). After excluding network failures, the final subset is 183 items. Its representativeness is validated in Q1 (Tables 1–3). All analyses will be included in the camera-ready.
>
> ### Q4: for threshold weakness
>
> > The 0.8 threshold lacks justification and sensitivity analysis, and its impact on rankings is unclear.
>
> We thank the reviewer and will include a full sensitivity analysis in the camera-ready version.
>
> The choice of threshold is informed by both theoretical and empirical considerations, with the final setting selected based on optimal experimental results (details are provided in our response to Reviewer `vah4`, Q2). We also analyzed the effect of different thresholds on model rankings, as discussed in our response to Reviewer `bziQ`, Q2, and found that rankings remain stable across thresholds.

---

> > ### Author Rebuttal · Reviewer_PXn5 · 2026-04-03
> >
> > The rebuttal adequately complements the paper and addresses my concerns.

---

> > > ### Author Response · Authors · 2026-04-03
> > >
> > > We are glad we were able to address your concerns. We would greatly appreciate your consideration of whether an updated score might be warranted.

---

### Decision · Program_Chairs · 2026-04-30

**Decision:**

Accept (spotlight)

**Comment:**

Most code generation benchmarks focus on correctness/output evaluation. But for scenarios like HTML/App generation, there is no right answer, and what matters is whether the intended functionality works while following real-world principles (physics, game rules, etc.). The benchmark contains 500 tasks, and comes with an evaluation framework MiniAppEval which launches the app in a real browser via Playwright, interacts with it, and generates traces. An LLM judge then reads these traces to score whether the functionality works. All reviewers felt:

  1. Interaction and functionality evaluation is an important problem for HTML app generation.
  2. MiniAppEval will be useful to drive progress in this area.
  3. The benchmark is hard. The best model achieves only 45.5% pass rate.
  4. Strong human agreement.
  5. The rebuttal addressed all existing concerns, with one reviewer explicitly confirming satisfaction.

Limitations:
Only 15.4% of the 500 tasks are real user queries, the rest are LLM-augmented variants.